# Brain connectivity changes underlying depression and fatigue in relapsing-remitting multiple sclerosis: A systematic review

Agniete Kampaite[1,2☯], Rebecka Gustafsson[1☯], Elizabeth N. York[1,2,3], Peter Foley[1,3], Niall J. J. MacDougall[3,4], Mark E. Bastin[1,2], Siddharthan Chandran[1,3,5], Adam D. Waldman[1,2‡]*, Rozanna Meijboom[1,2‡]*

**1** Centre for Clinical Brain Sciences, University of Edinburgh, Edinburgh, United Kingdom, **2** Edinburgh Imaging, Edinburgh Imaging Facility, University of Edinburgh, Edinburgh, United Kingdom, **3** Anne Rowling Regenerative Neurology Clinic, University of Edinburgh, Edinburgh, United Kingdom, **4** Department of Neurology, Institute of Neurological Sciences, Queen Elizabeth University Hospital, Glasgow, United Kingdom, **5** UK Dementia Research Institute, University of Edinburgh, Edinburgh, United Kingdom

☯ These authors contributed equally to this work.
‡ ADW and RM also contributed equally to this work.
* rozanna.meijboom@ed.ac.uk (RM); Adam.Waldman@ed.ac.uk (ADW)

**Data Availability Statement:** All relevant data are within the manuscript and its Supporting Information files.

## Abstract

Multiple Sclerosis (MS) is an autoimmune disease affecting the central nervous system, characterised by neuroinflammation and neurodegeneration. Fatigue and depression are common, debilitating, and intertwined symptoms in people with relapsing-remitting MS (pwRRMS). An increased understanding of brain changes and mechanisms underlying fatigue and depression in RRMS could lead to more effective interventions and enhancement of quality of life. To elucidate the relationship between depression and fatigue and brain connectivity in pwRRMS we conducted a systematic review. Searched databases were PubMed, Web-of-Science and Scopus. Inclusion criteria were: studied participants with RRMS ($n \geq 20$; $\geq 18$ years old) and differentiated between MS subtypes; published between 2001-01-01 and 2023-01-18; used fatigue and depression assessments validated for MS; included brain structural, functional magnetic resonance imaging (fMRI) or diffusion MRI (dMRI). Sixty studies met the criteria: 18 dMRI (15 fatigue, 5 depression) and 22 fMRI (20 fatigue, 5 depression) studies. The literature was heterogeneous; half of studies reported no correlation between brain connectivity measures and fatigue or depression. Positive findings showed that abnormal cortico-limbic structural and functional connectivity was associated with depression. Fatigue was linked to connectivity measures in cortico-thalamic-basal-ganglial networks. Additionally, both depression and fatigue were related to altered cingulum structural connectivity, and functional connectivity involving thalamus, cerebellum, frontal lobe, ventral tegmental area, striatum, default mode and attention networks, and supramarginal, precentral, and postcentral gyri. Qualitative analysis suggests structural and functional connectivity changes, possibly due to axonal and/or myelin loss, in the cortico-thalamic-basal-ganglial and cortico-limbic network may underlie fatigue and depression in pwRRMS, respectively, but the overall results were inconclusive, possibly explained by heterogeneity and limited number of studies. This highlights the need for further studies

**Funding:** "Funding for authors came from the MS Society Edinburgh Centre for MS Research (grant reference 133; AK, RM), Chief Scientist Office – SPRINT MND/MS program (grant reference MMPP/01; ENY), the Anne Rowling Regenerative Neurology Clinic (ENY), and the UK Dementia Research Institute which receives its funding from UK DRI Ltd, funded by the UK Medical Research Council, Alzheimer's Society and Alzheimer's Research UK (SC).".

**Competing interests:** The authors have declared that no competing interests exist.

including advanced MRI to detect more subtle brain changes in association with depression and fatigue. Future studies using optimised imaging protocols and validated depression and fatigue measures are required to clarify the substrates underlying these symptoms in pwRRMS.

# 1. Introduction

## 1.1 Multiple sclerosis

Multiple sclerosis (MS) is a chronic neuroinflammatory and neurodegenerative disease, with 2.3 million people diagnosed worldwide [1]. Central nervous system (CNS) damage in MS is typically characterised by white matter lesions (WMLs) in the brain and/or spinal cord, which are visible on magnetic resonance imaging (MRI), although atrophy is also recognised as an important feature [2]. Relapsing-remitting MS (RRMS) is the most common subtype (around 85% of cases) and is characterised by alternating periods of neurological dysfunction (relapses) and relative clinical stability (remissions) [3, 4]. RRMS presents with a wide range of features, including motor, visual, balance and sensory impairment [3]. Importantly, in addition to the more obvious physical manifestations of MS, 'hidden disability' such as fatigue and depression, affects most patients, is debilitating, and challenging to treat [5–7].

## 1.2 Depression and fatigue in MS

Higher prevalence of depression in MS than in the general population has been previously reported [8], and fatigue may affect 60–80% of people with newly diagnosed MS [9]. Both fatigue and depression are associated with decreased quality of life in people with MS [10] and are considered major debilitating symptoms [11], together affecting more than 50% of people with MS [10]. The relationship between depression and fatigue is complex; although considered distinct entities, there is a high degree of comorbidity and their phenotypes overlap (e.g., anhedonia, sleep disturbance) [12, 13]. Fatigue is considered both a symptom and a consequence of depression, and conversely, people with fatigue are more likely to report depressive symptoms [13, 14]. Associations of fatigue and depression and other MS symptoms, such as pain, cognition, and anxiety have also been found [15–21]. In view of the strong overlap of fatigue and depression, however, this review will focus on establishing a better understanding of the substrate for fatigue and depression, and their relationship to known MS pathobiology.

Depression is one of the most common psychiatric disorders, defined by depressed mood and/or loss of interest or pleasure [22]. Other symptoms are significant weight and appetite changes; reduction of physical movement; fatigue or loss of energy; negative self-image; reduced concentration; and suicidal thoughts [22]. There are various potential causes of depression, ranging from predisposing temperament and personality traits, exposure to traumatic and stressful life events, to genetic susceptibility [23, 24]. Multiple assessment tools are available for reliably measuring depression, some of which have been specifically validated for use in MS [25]. Depression is considered a co-morbidity of MS [7] and may be caused by reduced quality of life [26], including changes in mental wellbeing due to living with MS, side effects of medications, individual situations, and social circumstances [27]. Some studies, however, suggest that MS-specific pathophysiology, i.e., atrophy and inflammation of the CNS, contribute to high prevalence of depression in MS patients [28, 29]. This is supported by the observation that depression may be more prevalent in MS than in other neurodegenerative/inflammatory disorders [30–34]. There is, however, no correlation between depression and level of disability or disease duration in RRMS [35].

Fatigue is a complex and ambiguous symptom. Not only is it considered both a symptom and a consequence of depression [14], but it is also associated with numerous other physical and psychiatric diagnoses, due to its broad physical, cognitive, and emotional components [13]. Fatigue can appear spontaneously, or be brought on by a combination of internal or external factors, such as mental or physical activity, heat sensitivity, humidity, acute infection, and food ingestion [7, 36]. Commonly suggested primary mechanisms of fatigue in MS involve the immune system or damage to the CNS, such as inflammatory processes (e.g., cytokines), endocrine dysregulation, axonal loss, demyelination, as well as functional connectivity changes [9, 37, 38]. This review will focus on structural damage of the CNS in the white (WM) and grey matter (GM), specifically changes in structural and functional brain connectivity, as potential underlying mechanism of fatigue in pwRRMS.

Fatigue is difficult to define, but it has been described as "reversible motor and cognitive impairment, with reduced motivation and desire to rest" [39] or "a subjective lack of physical and/or mental energy that is perceived by the individual or caregiver to interfere with usual or desired activity" [40]. A distinction is made between performance fatigue (or fatigability) and subjective (or perceived) fatigue, where performance fatigability occurs through repeated activities and can be measured through assessments capturing functional decline [41, 42]. Subjective fatigue, on the other hand, is internally (and subjectively) perceived or experienced by an individual [41]. As subjective fatigue is a core symptom in people with MS [40], we will focus on this type of fatigue in the current review.

Measurement of subjective fatigue can prove difficult. A variety of fatigue scales are available—some of which are validated in MS [43, 44]—although a 'gold standard' has not been established [9]. Some of these measures consider subjective fatigue as one concept (e.g., fatigue severity scale [FSS] [45]), where others (e.g., fatigue scale for motor and cognitive functions [FSMC] [46]) differentiate between cognitive fatigue (e.g., concentration, memory, decision making) and motor fatigue (stamina, muscle strength, physical energy). In MS, fatigue is categorised as primary (caused by neurological abnormalities) and secondary (resulting from MS symptomatology) [9, 47]. The pathophysiology underlying primary MS fatigue is not yet clear [48], although previous studies have suggested overlapping brain abnormalities between fatigue and depression in MS [49, 50], which is unsurprising given their strong association [51].

Treatments for depression and fatigue in MS are limited, and there is some controversy regarding their efficacy [9, 52, 53]. Currently, few treatments (i.e., Amantadine, Modafinil, and selective serotonin reuptake inhibitors) are available in the UK for fatigue-specific management in MS [53]. However, a randomised, placebo-controlled, crossover, double-blind trial suggests that Amantadine and Modafinil are not better than placebo in improving MS fatigue and have more side effects [54]. Additionally, antidepressants, cognitive behavioural therapy [6] and cryotherapy [55] have had some success in reducing both depression and fatigue symptomatology in MS. Given the limited treatment success, underlying CNS changes of fatigue and depression in MS need to be elucidated, which may aid development of more effective targeted treatments for both symptoms in MS.

## 1.3 Magnetic resonance imaging in MS

MRI allows for non-invasive, *in vivo*, detection of underlying CNS damage in MS. MRI is sensitive to MS brain pathology, as shown by previous research [56]. Conventional ('structural') MRI has been widely used to study brain abnormalities in people with RRMS (pwRRMS) and provides information on location and severity of structural tissue damage such as WML burden and atrophy [57, 58], through qualitative reads or volumetric analyses. However, the

ability of conventional MRI to explain clinical symptomatology is limited [59], and evidence for a relationship between fatigue or depression and conventional MRI measures in mixed subtype MS is inconsistent [60, 61]. Advanced techniques, such as diffusion MRI (dMRI) and functional MRI (fMRI), can be used to investigate the role of more subtle brain abnormalities in the development of clinical symptoms in MS.

**1.3.1 Brain connectivity measures.** Diffusion MRI and fMRI can be used to study how different regions of the brain are connected, in terms of structure and function respectively, and form brain networks [62, 63]. In MS, damage to tissue microstructure (e.g., myelin and axons) is a core pathology even in early disease [64, 65]. Both intact myelin and axons are essential for signal transfer in the brain and thus successful functioning of brain networks [66]. Damage to brain microstructure directly impacts structural connectivity and may also change functional connectivity [67]. Brain connectivity abnormalities likely result in clinical symptomatology and may be underlying of MS symptoms such as fatigue and depression [68, 69].

**1.3.2 Diffusion MRI.** Diffusion MRI is sensitive to occult tissue damage at a microstructural level, which cannot be detected by conventional MRI [70], and allows for studying structural brain connectivity. A widely used dMRI model is diffusion tensor imaging (DTI) [71]. DTI uses brain water molecule displacement to estimate the organisation of WM tracts and tissues at the microstructural level [72]. DTI metrics, such as fractional anisotropy (FA) and mean diffusivity (MD), are sensitive to changes in this microstructure, and are thought to reflect myelin and axonal damage [70, 72]. Decreases in FA and increases in MD in several WM tracts have been linked to clinical disability as well as fatigue and depression scores in people with MS [61, 73]. More recently, a DTI marker called 'peak width of skeletonized mean diffusivity (PSMD) [74] was proposed to reconstruct microstructural WM damage across the brain and provide a global measure of structural connectivity [75, 76]. A newer dMRI model is neurite orientation dispersion and density imaging (NODDI), which allows for more specific characterisation of WM microstructure than DTI, i.e., neurite (axon and dendrite) density, and dispersion of neurite orientation [77]. Previous studies using NODDI have shown that neurite density is affected in MS [65, 78, 79].

**1.3.3 Functional MRI.** Functional MRI provides an indirect measure of brain activity and functional connectivity, using the blood oxygenated level-dependent (BOLD) technique, which reflects changes in blood oxygenation, volume, and flow [80]. Task-based fMRI can be used to identify brain activation in regions simultaneously involved in task performance, whereas resting-state fMRI (rs-fMRI) is used to explore intrinsic functional connectivity between areas of the brain, known as resting-state networks (i.e., default mode network, salient network, basal ganglia network), based on coherence of spontaneous fluctuations in BOLD signal [81–83]. Previous literature has shown brain activity and functional connectivity changes in the frontal lobe, limbic system and basal ganglia linked to high cognitive fatigue [80, 84] and depression [85] in individuals with MS. Additionally, functional connectivity changes in the default mode network (DMN), comprising mainly the medial prefrontal cortex, precuneus, posterior cingulate gyrus and inferior parietal gyrus [86, 87], have been associated with cognitive impairment and depression in people with MS [88, 89]. The sensorimotor network (SMN), including postcentral and precentral gyri and the supplementary motor area (SMA), has also been suggested to show changes in functional connectivity associated with fatigue in MS [90, 91].

## 1.4 Purpose

Previous systematic reviews concluded that abnormalities of the cortico-striato-thalamo-cortical loop underlie fatigue symptomatology in MS of varying subtypes [61, 92, 93]. Moreover,

depression severity in MS is associated with structural and fMRI changes in several brain regions, of which frontal and temporal lobes are the most common finding [5, 94]. Brain connectivity changes underlying depression, fatigue, or both, specific to pwRRMS have not, however, been reviewed. The dominant pathophysiological processes and relapsing-remitting clinical features in RRMS differ from progressive MS subtypes, and it is therefore important to study underlying brain changes related to fatigue and depression, specifically in this group. Moreover, to our knowledge, potential overlap of brain connectivity changes underlying depression and fatigue in pwRRMS have not previously been reviewed systematically.

The aim of this study is to systematically review the literature to elucidate the relationship between structural and brain connectivity MRI measures and depression or fatigue in pwRRMS. This may provide new insights into axonal and/or myelin changes in RRMS related to depression and fatigue.

## 2. Methods

Ethics committee approval was not required for the current review.

The work was focussed on topics that have previously been identified as major priorities for pwMS [95, 96].

### 2.1 Inclusion and exclusion criteria

A systematic review of published primary research articles on brain abnormalities measured with structural, diffusion or functional MRI and their associations with fatigue or depression in pwRRMS was conducted. Preferred Reporting Items for Systematic Reviews and Meta-Analyses (PRISMA) guidelines [97] were followed where possible (see S1 and S2 Checklists for the PRISMA checklist). Studies were included if they met the following inclusion criteria: (1) structural, diffusion or functional MRI was used to study brain changes, (2) included a minimum sample size of 20 participants, (3) assessed either RRMS alone or distinguished between MS subtypes, and (4) used depression or fatigue assessments validated for use in MS, based on three previous reviews of MS-related depression and fatigue [5, 61, 94] (Depression assessment tools: Beck Depression Index (BDI) [82], Beck Depression Index-II (BDI-II) [83], Diagnostic and Statistical Manual V semi-structured interview (DSM-V) [84], Centre for Epidemiological Studies–Depression (CES-D) [84], Chicago Multiscale Depression Inventory (CMDI) [84], Patient Health Questionare-9 (PHQ-9) [84], Hospital Anxiety and Depression Scale (HADS) [87], Hamilton Depression Rating Scale (HDRS) [88]; Fatigue assessment tools: Fatigue Severity Scale (FSS) [29], Modified Fatigue Impact Scale (MFIS) [29], Fatigue Impact Scale (FIS) [85], Fatigue Scale for Motor and Cognitive functions (FSMC) [31], Checklist of Individual Strength (CIS-20r) [86]. Short descriptions for each measure can be found in Gümüş [85] or Cheung [89]). Studies were excluded if: (1) they did not distinguish between subjects with RRMS and other MS subtypes in their results and data analysis, (2) if the participants were under the age of 18, or (3) if they assessed the effects of disease modifying therapies (DMTs) on MRI or clinical measures (unless they controlled for DMT usage).

### 2.2 Search strategy and selection process

The literature search was conducted by two independent reviewers using three online databases: PubMed, Web-of-Science and Scopus, and considered publications up to 18-01-2023. The databases were searched using a title, abstract and keyword search, for publications written in English and published in the past 22 years (2001–2023). The following search terms were used: '*fatigue*' or '*depression*' or '*depressive symptoms*', in combination with '*relapsing-remitting multiple sclerosis*' or '*relapsing remitting multiple sclerosis*', in combination with

'*magnetic resonance imaging*' or '*MRI*' or '*neuroimaging*' or '*brain atrophy*' or '*diffusion tensor imaging*' or '*diffusion MRI*' or '*dMRI*' or '*NODDI*' or '*neurite orientation dispersion and density imaging*' or '*functional MRI*' or '*fMRI*' or '*resting state*'. After duplicates were excluded, publication titles and abstracts were read by two independent reviewers and any studies clearly not meeting inclusion criteria were excluded. In case the abstract lacked sufficient information, a brief read of the paper was performed. The remaining studies were then read in full, and further articles were excluded using the criteria described in section 2.1 (Fig 1). Final selections were compared to reach consensus. In case of a disagreement, the reviewers re-read the paper and either amended their decision or made further arguments for their initial choice. Persisting discrepancies were discussed together with a third reviewer and final decisions were made by consensus. The data was extracted by one reviewer into a standardised table designed for this review (S2 and S7 Tables).

## 2.3 Analysis approach

Outcome measures comprised pre-specified structural, diffusion and functional MRI measures. Structural measures included regional and global brain, WM, and GM volume, WML volume, global and regional lesion count, and brain parenchymal fraction (BPF). Diffusion measures included DTI-derived whole-brain, regional and tract-specific FA, MD, axial diffusivity (AD), and radial diffusivity (RD); as well as regional and tract-specific NODDI, and PSMD metrics. For fMRI, both task-based and resting-state fMRI measures were included.

A qualitative approach was used to summarise the observations in the identified studies, due to heterogeneity in outcome measures, population, and experimental approach. The number of comparable experimental designs was too small to perform meaningful quantitative meta-analysis. For transparency, all details about included studies and statistically significant results are summarised in S7 Table and Table 1, respectively. Findings of no significant association between the examined clinical and MRI imaging variables are summarised in S8 Table.

## 2.4 Quality assessment

Institute of Health Economics (IHE) Quality Appraisal of Case Series Studies Checklist was used to assess the quality of the longitudinal studies included [98] and the 'Appraisal tool for cross sectional studies' (AXIS) was used to assess quality of cross-sectional studies [99]. Two reviewers conducted the quality assessment independently. See the full overview of the quality assessment process in the Supporting Information.

## 3. Results

### 3.1 Literature search and study characteristics

The initial database search (Fig 1) identified 604 candidate publications of which 60 studies met the inclusion criteria (Table 1). Eleven out of these 60 studies investigated the associations between depression and MRI measures [29, 50, 100–106], 41/60 assessed fatigue in association with MRI outcomes [35, 70, 107–135], and 8/50 investigated both depression and fatigue in association with MRI measures [136–139]. Substantially fewer papers examining associations between CNS abnormalities and depression met the inclusion criteria, with five studies using DTI and five using fMRI measures. Of note, we found very few studies that used NODDI or PSMD and none of them met the inclusion criteria. See S1 Table for an overview of all studies reviewed and their selection process.

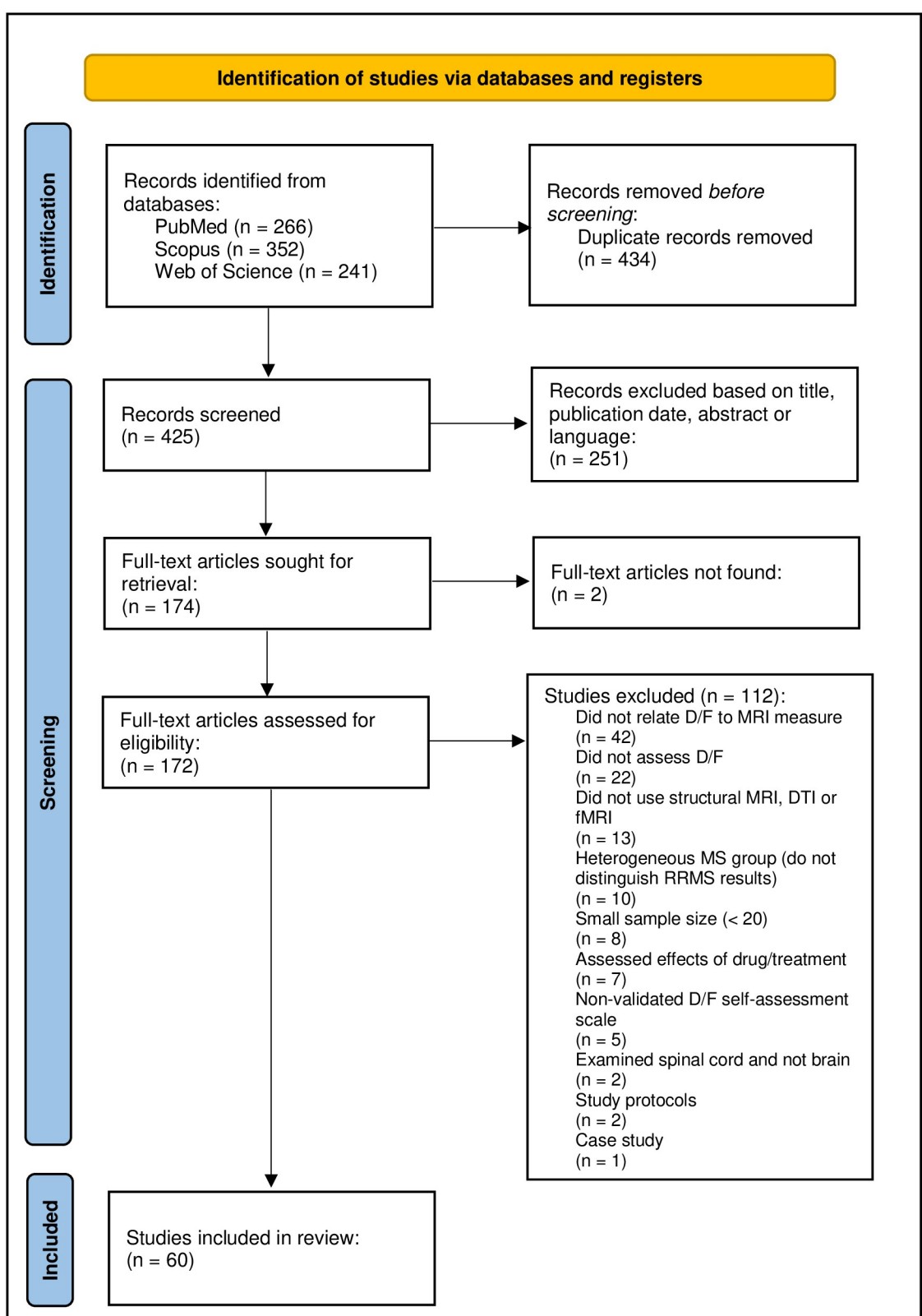

**Fig 1. Flowchart of literature search.** Performed in January 2023. Based on PRISMA 2020 flow diagram for new systematic reviews which included searches of databases and registers only [97]. D: Depression, DTI: Diffusion tensor imaging, F: Fatigue, (f)MRI: (functional) magnetic resonance imaging, (RR)MS: (relapsing-remitting) multiple sclerosis.

**Table 1. Overview of study characteristics and findings for included publications (N = 60) in the current systematic review.**

| | Authors | Fatigue assessment | Depression assessment | MRI sequence | Structural MRI | Diffusion MRI | Functional MRI | Major findings |
|---|---|---|---|---|---|---|---|---|
| **Depression** | Benesova et al. [100] | N/A | HDRS ≥ 18; 3 patients had severe, 2 moderate, 5 light D | T1W, T2W | Whole brain Lesion area | N/A | N/A | Bigger lesion area in frontal lobe in patients with D compared to nD. |
| | Nygaard et al. [50] | FSS: mean(SD) 4.1(1.7); Min-max 1–7; 49.2% patients had FSS > 4 | BDI: mean(SD) RRMS: 8.4(5.9); HC: 3.9(4.0); Min-max: RRMS: 0–24; HC: 0–16; % BDI >12 —RRMS: 27.1; HC: 6.9; excluded BDI > 16 | T1W, FLAIR | cortical surface area, thickness, and volume | N/A | N/A | D was associated with smaller cortical surface area of frontal pole, pars orbitalis, and orbital frontal region; the rostral and caudal middle frontal and the pre- and post-central regions; L-middle temporal, L-fusiform and L-parahippocampal regions. D was associated with volumes of orbital frontal and pars orbitalis, the superior frontal, rostral and caudal middle frontal, pre- and post-central regions; R-supramarginal and R-superior temporal regions; L-fusiform and L-inferior temporal region. Higher fatigue scores were associated with smaller cortical volumes in the rostral and caudal middle frontal, and in parts of the pre- and post-central regions, of the right hemisphere. |
| | Rojas et al. [29] | FIS: nD 12 ± 5; D 14.7 ± 4 | BDI-II, DSM-V | T1W, T2W, FLAIR, DTI | Total brain, GM and WM volumes, neocortical GM and T2 lesion volumes | Mean FA in cortico-spinal tract | N/A | Negative correlation between BDI-II scores and total brain volume and neocortical GM volume. Positive correlation with T2 lesion volume. No differences in global FA and in WM volume. |
| | Yaldizli et al. [105] | FSMC cognition: All MS patients 27.3 (10.4); RRMS 25.9(10.8); SPMS 31.2(8.6); PPMS 26.9(10.8). FSMC physical: All MS patients 30.9(11.1); RRMS 28.5(11.5); SPMS 37.5(6.9); PPMS 31.7(8.6) | CES-D All MS patients 12.8 (10.5) RRMS 11.6(10.1) SPMS 15.7(11.2) PPMS 16.4(10.1) | T1W, PD/T1W | Olfactory bulb volume, total WM lesion load, BPF | N/A | N/A | No significant correlations |
| | Gold et al. [101] | NA | BDI-II ≥ 13 HC: 1.6+.5; RRMS: 9.5 +1.6. | T1W, T2W, FLAIR | Hippocampus Volume | N/A | NA | Reduced volume of CA23DG region in patients with D compared to nD. |
| | Štecková et al. [104] | N/A | BDI-II mean±SD (range): CIS: 9.8±13.2 (0–54); MS5:14.0±11.5 (2–47); MS10:11.3±8.8 (1–24); 17/43 patients were taking SSRI. | T1W | Thalamus volume | N/A | N/A | High BDI scores correlated with reduced volume of thalamus five years after diagnosis and increased volume ten years after diagnosis. |
| | Nigro et al. [102] | FSS: range: nD 2.7 ± 1.4(1–5.6); D 5.3 ± 1.4 (1.4–7.6) | DSM-V, BDI-II | T1W, FLAIR, DTI | GM, WM volumes, lesion load | Tractography WM connectivity | N/A | No structural correlates to BDI scores. Increased shortest path length between R-hippocampus and R-amygdala; dorsomedial and ventrolateral PFC and the occipitofrontal cortex |
| | Carotenuto et al. [106] | N/A | HDRS: mean±SD RRMS 6.39±4.21; HC 0.56±0.96 | T1W, T2W, FLAIR, rs-fMRI | Whole brain lesion volume | N/A | rs-fMRI. ROI: serotonergic, noradrenergic, cholinergic, and dopaminergic networks | No structural correlates to HDRS scores. Reduction in serotonergic and noradrenergic activity as well as increased cholinergic activity was positively correlated with high HDRS scores. |
| | Riccelli et al. [103] | FSS: [range] RRMS 3.62 (1.83) [1–7]; | DSM-V; BDI [range]: HC 7.35 (5.08) [0–16] RRMS 11.10 (9.92) [0–45]; BDI-Fast Screen [range]: HC 1.94 (1.52) [0–5] RRMS 3.12 (3.55) [0–16]; BDI-Somatic Subscale [range]: HC 5.16 (3.38) [0–10] RRMS 7.64 (6.15) [0–27] | T1W, FLAIR, fMRI: categorise emotions of grey-scale photographs of faces | Total lesion load | N/A | fMRI ROI: Prefrontal cortex, orbitofrontal cortex, cingulate, amygdala, hippocampus | Decreased activation of R-subgenual cingulate cortex in participants with high BDI scores. Negative correlation between BDI scores and functional connectivity between L-hippocampus with L-orbitofrontal cortex, R/L-DLPFC and R-amygdala. |
| | Hassan et al. [140] | N/A | DSM-V; BDI: 9–29 | T1W, T2W, FLAIR, DTI | N/A | FA in Limbic system | N/A | D patients showed decreased FA values in the cingulum, UF, and the fornix; no differences in the mean FA of the anterior thalamic radiations; compared to HC, significant reduction in the mean FA of the cingulum, UF and the fornix; D had reduced FA of the cingulum, uncinate fasciculus and the fornix. No significant difference was found between the FA values of the anterior thalamic radiations in both groups. No conventional MRI reported. |
| | Kopchak and Odintsova [141] | NA | BDI, no data | No data | brain volume; bifrontal, caudal, ventricular indices; width of lateral ventricle in coronal parasagittal planes, diameter of 3rd and 4th ventricles in coronal plane, subarachnoid spaces (cranio-cortical width, sino-cortical, interhemispheric width), GM (cortical) thickness | NA | NA | Combined lesions of frontal lobe and corpus callosum, fronto-temporal region were associated with depression. |

*(Continued)*

**Table 1.** (Continued)

| | Authors | Fatigue assessment | Depression assessment | MRI sequence | Structural MRI | Diffusion MRI | Functional MRI | Major findings |
|---|---|---|---|---|---|---|---|---|
| F and D | Hildebrandt et al. [137] | FSS 23 patients had ≥ 4.5 (51% of the total group) | BDI: 24% of the patients with some symptoms of a depression (BDS≥ 16) and 7% with a definite depression (BDS≥20). | T1W | BPF, ventricular brain fraction | N/A | N/A | Ventricular brain fraction or BPF did not correlate with either BDI or FSS scores. |
| | Hildebrandt and Eling [136] | FSS F: FSS>5, nF: FSS≤5 Change in F (Mean[SD]): No increase: 5.4 [1.5]; Increase: 3.1 [1.6]; Change in depressive mood (Mean [SD]): No increase: 35.8 [13.9]; Increase: 38.5 [19.4] | BDI>12 considered depressed. Excluded the items on tiredness and sleep disorders. No data reported | T1W | BPF | N/A | N/A | No significant correlations |
| | Lazzarotto et al. [139] | F: FSS ≥ 5 mild-moderate (2–5) and severe (≥5); D n = 16 (2.44–6.55) mean 4.43 SD 1.28; nD n = 45 (1–5.56) mean 2.64 SD 1.38 | D: BDI-II ≥ 14 only stats data reported | T1W, FLAIR, DIR | Cerebellum and brainstem Volume | N/A | N/A | BDI correlated with lower volume of R-vermis crus I and FSS with cerebellar lobule right V atrophy. |
| | Jaeger et al. [138] | FSS: fatigued if they had at least a score of FSS = 4. FSS Median (IQR): F: 5.2 (1.3); nF: 2.6 (1.9); HC: 1.9 (1.3) | BDI-II ≥ 20 excluded BDI-II Median (IQR): F: 11 (7); nF: 3.5 (5.5); HC: 2 (4) | T1W, FLAIR, rs-fMRI | Volumes | N/A | rs-fMRI. ROI: caudate nucleus, putamen | No correlation between macrostructural volumes and fatigue. Negative correlation between rs-FC and BDI-II scores: L-ventral striatum and L-postcentral gyrus/R-precentral gyrus. BDI scores did not correlate with functional connectivity for other striatum subregions, the whole caudate and putamen and the DLPFC. Negative correlation between rs-FC and FSS scores: L-ventral striatum and R-precentral gyrus; R-ventral striatum and R-postcentral gyrus, superior ventral striatum and VTA, inferior temporal gyrus, SMA, attention and reward networks, parietal lobule, middle frontal gyrus, caudate and SMA. Positive correlations with FSS scores: L-DLPFC and L-parietal operculum; R-DLPFC and L-anterior supramarginal gyrus; R-parietal operculum, L-pre/postcentral gyrus and R-anterior supramarginal gyrus. |
| | Golde et al. [142] | FSMC total: RRMS 50.00 ± 21.84; HC 27.57 ± 6.76; FSMC cognitive RRMS 25.20 ± 11.50; HC 13.83 ± 3.90; FSMC motor: RRMS 24.80 ± 10.92; HC 13.73 ± 3.25 | HADS-D depression: RRMS 2.43 ± 2.90 HC 1.60 ± 2.31 | BOLD, MP-RAGE, DTI | whole and regional (hippocampus, amygdala, fusiform gyrus)) brain and GM volumes | DTI | rs-fMRI | No correlation with whole-brain volume or structural connectivity measures. F scores correlated positively with FFG based FC to the MPFC and negatively with FFG-based FC to right lateral PFC; F scores were not significantly correlated with hippocampus-based FC across the brain. Fusiform gyrus-based FC correlated with fatigue. |
| | Beaudoin et al. [143] | FIS (mean): 83.96 ± 30.08. 50% patients—low level F, 29%—moderate level F, 21%—high level F | BDI-II: (mean): 11.13 ± 10.85. 67% patients—normal score, 21% mild D, 3 cases (12%)—severe D | T1W, T2W, HARDI | NA | Total apparent fiber density (AFDtot) and number of fiber orientation (NuFO), WM bundles volumes, total lesion load, lesion volume in bundles, tractometry | NA | BDI-II was associated with diffusion abnormalities in the R superior longitudinal fasciculus. A decrease in FA, an increase in RDt and MDt were associated with a higher level of depressive symptoms. Free water fraction and HARDI derived measures (AFDtot, NuFO) were not associated with the clinical results. Furthermore, no correlation was found between the neurocognitive testing results and the global brain lesion load. When the WM bundles were studied individually, their respective proportion of lesioned tissue was also not associated with the clinical data. |
| | Kever et al. [144] | FSS, MFIS no data | BDI-II no data | T1W, FLAIR, T2W | Network structure, total brain, GM, WM volumes, T2LV, subcortical GM structures (bilateral amygdala, caudate, nucleus accumbens, putamen, pallidus, hippocampus, and thalamus) | NA | NA | No relationships of network structure to depression or fatigue were found. |
| | Romanello et al. [145] | FSS (mean ± SD): (MS EDSS ≤ 1) – 0.63 ± 0.68; (MS EDSS ≥ 2) 0.58 ± 0.87; | BDI-II (mean ± SD): (MS EDSS ≤ 1) – 0.37 ± 0.75; (MS EDSS ≥ 2) 0.32 ± 1.06 | T2W, rs-fMRI | total brain atrophy and lesion load | NA | FC | D severity was positively related to functional dynamics as measured by state-specific global connectivity and DMN connectivity with attention networks, while F was related to reduced frontoparietal network connectivity with the basal ganglia. |

*(Continued)*

**Table 1.** (Continued)

| | Authors | Fatigue assessment | Depression assessment | MRI sequence | Structural MRI | Diffusion MRI | Functional MRI | Major findings |
|---|---|---|---|---|---|---|---|---|
| Fatigue | Cavallari et al. [111] | MFIS median (range) all 23 (0–67) C 37 (5–67) non-C 13 (0–42); MFIS cognitive, median (range): All patients 11 (0–31); Converters 13 (1–31); non-converters 6 (0–20) (n = 33) | CES-D score ≥16 is considered in the depressed range | T1W | BPF, total T2 lesion volume | N/A | N/A | No significant correlations |
| | Tomasevic et al. [120] | F: MFIS ≥ 16 nF: MFIS < 15 FSS: mean (sd) 3.6 (1.8); MFIS tot: mean (sd) 26.6 (13.8); | BDI-II > 13 excluded mean (sd) 7.2 (3.9) | T1W GD+, T2W, T1W, FLAIR | Whole brain, Thalamus volume, cortical thickness | N/A | N/A | No significant correlations |
| | Morgante et al. [132] | F: FSS > 4 nF: <4 mean FSS: nF 2.2 ± 0.9; F 4.9 ± 0.8 | HDRS (mean+?): nF 6.1 ± 4.6; F 6.4 ± 4.8 | T1W GD+, T1W, transcranial magnetic stimulation | Lesion load | N/A | N/A | No significant correlations between lesion volume and fatigue |
| | Téllez et al. [134] | F: FSS ≥ 5 and/or MFIS> 38; nF: FSS <4.0. means (SD) FSS: RRMS 4.8 (1.5); HC 3.2 (1.2); means (SD) MFIS: RRMS 35.2 (22.9) HC 18.3 (11.9); means (SD) FSS: F 5.9 (0.7); nF 3.6 (1.15); means (SD) MFIS: F 48.0 (20.6); nF 18.7 (12.4); | BDI BDI (means (SD)): RRMS 9.8 (9.1) HC 5.2 (3.4); F 12.2 (8.9); nF 5.6 (6.4) | T2W, proton magnetic resonance | Frontal WM, lentiform nucleus, lesion load | N/A | N/A | No significant correlations between lesion volume and fatigue. |
| | Yarraguntla et al. [135] | LF: FSS ≤ 3 MF: FSS 3.1–5 HF: FSS ≥ 5.1 Mean(SEM) FSS: baseline: HF 6(0.12), MF 4(0.14), LF 1.89(0.2); Total 4.35(0.26), FSS range: HF (5.1–7), MF (3.1–5), LF (1–3), Total (1–7); Mean FSS at year 1: HF 5.8(0.26), MF 3.81 (0.42), LF 2.6(0.46), Total 4.19(1.3); FSS range: HF (3.7–7), MF (1.1–6.7), LF (1–6); Total (1–7) | Excluded participants with diagnosed clinical depression | T2W, FLAIR, MR spectroscopy | T2 lesion load, normal appearing WM | N/A | N/A | No significant correlations between T2 lesion volume and fatigue. |
| | *Altermatt et al. [107]* | *F: FSS ≥ 4 MFIS Median (range) 23 (0–63)* | *NA* | *T1W, T2W, FLAIR* | *Whole brain Lesion load* | *N/A* | *N/A* | *High FSS and MFIS correlated with increased lesion load in posterior corona radiata.* |
| | *Damasceno et al. [113]* | *FSS F: FSS score ≥ 4; FSS mean/SD RRMS 3.54 ± 1.65, HC 2.65 ± 0.88* | *Used our non-approved test* | *FLAIR, T1W, T1W* | *WM cortical lesions* | *N/A* | *N/A* | *Cerebellar cortical lesion volume was the only independent predictor of fatigue. Participants with F had higher GM lesion and GM volumes in cerebellum compared to nF. High FSS scores correlated with increased volume of thalamus, decreased volume of caudate and nucleus accumbens. No correlation to brain lesion volume nor to cortical and subcortical GM volumes.* |
| | *Calabrese et al. [110]* | *MFIS, FSS F: FSS ≥ 4 in all three examinations (baseline, 3 and 6 months); FSS mean: F 5.1(0.75) (range 4.00–6.67) nF: 2.2 (1.0) (range 0–3.88)* | *D: BDI ≥ 18* | *FLAIR, T1W* | *Thalamic and basal ganglia volume, regional cortical thickness* | *N/A* | *N/A* | *Lower volume of putamen, caudate nucleus, thalamus, superior frontal gyrus and inferior parietal gyrus in participants with F compared to nF.* |
| | *Yaldizli et al. [122]* | *FSS (mean and 95% CI) Total 3.37 ± 1.88 (2.92; 3.82) FSS<4: 2.1 ± 1.04 (1.78; 2.43) FSS≥4: 5.27 ± 1.09 (4.85; 5.69)* | *BDI ≥ 15 excluded* | *T1W GD+, T1W, T2W, FLAIR* | *Corpus callosum Volume* | *N/A* | *N/A* | *FSS correlated with the reduction in corpus callosum volume over 5-years compared to nF.* |
| | *Saberi et al. [131]* | *MFIS No scores* | *Excluded BDI-FS ≥ 10* | *T1W* | *Thalamus sub-region volume* | *N/A* | *N/A* | *Atrophy of left superior, anterior, and medial anterior thalamus was positively correlated with cognitive fatigue.* |
| | Niepel et al. [116] | MFIS, FSS F: FSS ≥ 5.0, nF: FSS ≤ 4. | N/A | 3D FLASH, T2W | Thalamus, putamen, caudate nucleus T2 lesion load | N/A | N/A | No structural differences between groups. |
| | Zellini et al. [124] | F: FSS ≥ 5 nF: FSS ≤ 4 | N/A | T1W, T2W | T2 lesion load | N/A | N/A | No structural differences between groups. |
| | Bisecco et al. [109] | F: FSS ≥ 4 Mean FSS (range): HC 2 (1–3.9); RRMS 3.6 (1–6.8); nF 2 (1–3.6); F 5.2 (4.2–6.8); | Excluded people with clinical depression Used our non-approved test | T1W, T2W, DTI | Whole brain volume (lesion, GM, WM) | FA, MD, RD | N/A | No structural- or diffusion differences between groups. |
| | Codella et al. [112] | F: FSS ≥ 4 Mean FSS (SD) nF 19.7 (5.2) F 38.9 (7.3) | N/A | DE TSE, 2D GE, pulsed gradient spin-echo echo-planar | magnetisation transfer ratio | MD | N/A | No structural- or diffusion differences between groups. |
| | *Andreasen et al. [108]* | *F: FSS ≥ 4; (median (range)): F: 6.3 (5–7); nF: 2.8 (1–4); HC: 2.7 (2–4);* | *Major Depression Inventory score ≥26 (not included in the study)* | *T1W, T2W, FLAIR, DTI, MRS proton spectroscopy* | *BPF, lesion load* | *FA, MD* | *N/A* | *High FSS negatively correlated with volume of: R-superior frontal, R-anterior cingulate, L-anterior frontal, R-middle temporal, R-superior temporal gyrus, L-anterior insula, R-superior parietal, R-inferior parietal, L-inferior parietal gyrus, and R-caudate nucleus. No diffusion correlates to FSS scores.* |
| | *Pardini et al. [117]* | *F: MFIS ≥ 37 (mean± SD): MFIS 27.6±17.3* | *NA* | *T1W, PD/T2W, DTI* | *Whole brain volume* | *FA* | *N/A* | *Significant association between structural damage and fatigue levels in two discrete white matter clusters in the left cingulate bundle. The damage in these clusters was associated with loss of structural connectivity in the anterior and medial cingulate cortices, dorsolateral prefrontal areas and in the left caudate. MFIS was associated with WM diffusion measures nearby to the anterior and medial cingulate cortices, respectively.* |

(Continued)

**Table 1.** (Continued)

| Authors | Fatigue assessment | Depression assessment | MRI sequence | Structural MRI | Diffusion MRI | Functional MRI | Major findings |
|---|---|---|---|---|---|---|---|
| Wilting et al. [121] | F: FSMC > 27 nF: FSMC < 22 FSMC (range) nF 33 (20–58) F 68 (51–97) | HADS > 10 excluded | T1W, FLAIR, DTI | GM, WM, and CSF fractions; BPF and lesion volume | MD, FA | N/A | Motor fatigue was weakly positively correlated with lesion volume and thalamic lesion load. Lesion volume was not correlated with cognitive fatigue. Higher MD and lower FA in the thalamus and basal ganglia (including the caudate nucleus, globus pallidus and putamen) in participants with cognitive F. |
| Pardini et al. [118] | MFIS mean(SEM) HF: MFIS > 38 LF: MFIS < 38 RRMS 31.1(18) HF 20.2(10) LF 51.4(9.9) | BDI ≥ 18 excluded | T1W, T2W, DTI | Cortical and deep WM ROI volumes | DTI | N/A | No structural measures associated with MFIS scores. High MFIS scores negatively correlated with structural connectivity of: internal capsule, forceps minor, anterior thalamic radiation, and cingulate bundle and inferior fronto-occipital fasciculus. MFIS scores negatively correlated with FA values in the deep left frontal WM. |
| Pokryszko-Dragan et al. [70] | FSS (Mean/SD): 4.4/1.7 | N/A | T1W, T2W, FLAIR, DWI, DTI, 3D-FSPGR GD+ | Volume ROI: corpus callosum, thalamus, cerebellar peduncles | MD, FA | N/A | No structural- or diffusion correlates with fatigue severity. |
| Yarraguntla et al. [123] | HF: FSS > 5 MF: FSS 3–5 LF: FSS 0–3 FSS mean(SEM) 4.35±.26 range (1–7) Mean FSS at year 1: 4.19±.3 range (1–7); Mean FSS at baseline: mean(SEM) (Range): HF: 6 ±.12(5.1–7); MF: 4±.14(3.1–5); LF:1.89±.2 (1–3); Total: 4.35±.26(1–7); Mean FSS at year 1: mean+SEM(Range): HF: 5.8±.26 (3.7–7); MF: 3.81±.42(1.1–6.7); LF: 2.6±.46 (1–6); Total: 4.19±.3(1–7). | Excluded people with clinical depression | T1W, DTI | Thalamus, pallidum, R-temporal cortex ROI, lesion load | MD, FA | N/A | High FSS negatively correlated with pallidum volume. Lower FA in R-temporal cortex in participants with HF compared to LF. |
| Zhou et al. [125] | MFIS-5 scores RRMS 11.15 (6–17) HC 0.25 (0.0–1.0) | N/A | T1W, T2W, DTI, rs-fMRI, FLAIR | Thalamic volume, total WM lesion load, BPF | MD, RD, structural connectivity | rs-fMRI: FC | Higher MD, RD, and AD in thalamocortical prefrontal WM tracts of individuals with high MFIS scores. No correlation between rs-fMRI and fatigue scores. |
| Finke et al. [114] | FSS Mean ±SEM(Range): RRMS: 3.94 ± 0.26(1.00–7.00); HC: 2.16 ± 0.19(2.16–4.67) | BDI ≥ 17 excluded | T1W, DTI, FLAIR, rs-fMRI | Whole brain Volume | FA, MD, parallel diffusivity and RD | rs-fMRI | No correlation between FSS or BDI scores and any structural or DTI measures. Negative correlations with FSS in rs-fMRI: R-pallidum and L/R-MPFC, L-precuneus and R-precuneus; L-putamen and L/R-MPFC, L-middle frontal gyrus and R/L-precuneus; L-MPFC and R-MPFC; R-caudate and L/R-MPFC. Positive correlations with FSS: caudate and precentral gyrus. |
| Filippi et al. [126] | F: FSS ≥ 25 nF: FSS <25 mean (SD) F: 39.5 (7.1); NF: 19.3 (5.2). | Montgomery and Asberg Depression Rating Scale > 16 excluded | T1W, fMRI, DE TSE | Lesion load, brain volume | N/A | fMRI: repetitive flexion and extension of fingers | No structural differences between groups. High FSS scores correlated with decreased FC between intraparietal sulcus, ipsilateral Rolandic operculum and thalamus. F patients had a more significant relative activation of the contralateral cingulate motor area, and reduced activation of cerebellum, precuneus and regions of the frontal lobe. |
| Huang et al. [115] | MFIS 17.0 ± 10.9—not clear what stats but probably mean | N/A | T1W, T2W, FLAIR, rs-fMRI | BPF, total WM lesion load | N/A | rs-fMRI | High MFIS correlated with disrupted connectivity in the R-superior temporal gyrus, and hypoactivity of DAN and DMN, increased connectivity in R-superior temporal gyrus and R-parahippocampal gyrus. |
| Rocca et al. [35] | F: MFIS≥38, nF: MFIS <38 Mean MFIS global (range) nF 22.5 (4–35) F 50.4 (38–71) Mean pMFIS (range)– 12.4 (2–30) 24.8 (14–33) Mean cMFIS (range)– 8.3 (0–18) 21.0 (9–34) Mean psMFIS (range)– 1.8 (0–4) 4.6 (2–8) | MADRS >9 excluded | T1W, T2W, fMRI | Lesion volume, normalised brain, WM, GM volumes | N/A | fMRI—finger-tapping test (using the average number of taps per 30s) and 9-hole peg test. | No difference in structural measures was found between F-MS and nF-MS patients. Increased middle frontal gyrus activity was related to MFIS scores. F-MS patients had reduced activations of the bilateral middle temporal gyrus, L-pre-SMA, L-SMA, bilateral superior frontal gyrus, L-postcentral gyrus, L-putamen, and bilateral caudate nucleus. F-MS patients experienced an increased recruitment of the bilateral putamen during the task. |
| Rocca et al. [133] | F: FSS ≥ 25 nF: mean [SD] 15.6 [4.8]; F: mean [SD] FSS = 37.7 [7.7] | MADRS: Excluded individuals with clinical D | T1W, T2W, fMRI | Whole brain | N/A | fMRI: kinematic coordinated hand/foot movements | No correlation between fatigue and lesion volume. RRMS with F had a higher activation of L-SII, R-precuneus, cerebellum and decreased activation of R-thalamus, R-basal ganglia, L-inferior frontal gyrus, and cingulate motor area compared to nF during movement. |

*(Continued)*

**Table 1.** (Continued)

| Authors | Fatigue assessment | Depression assessment | MRI sequence | Structural MRI | Diffusion MRI | Functional MRI | Major findings |
|---|---|---|---|---|---|---|---|
| Specogna et al. [128] | FSS F: FSS >5 nF:<4 | BDI > 16 excluded | T1W, T2W, T1W GD+, FLAIR, fMRI. | Lesion volume | N/A | fMRI: finger tapping against thumb | No significant difference in lesion burden. F patients had greater activation of the premotor area ipsilateral to the movement at the level of the right putamen and of the middle frontal gyrus on the right DLPFC. F group showed bilateral activation of the SMA and cerebellum. |
| Svolgaard et al. [119] | FSMC—Mean/Range/SD—RRMS 59.3 (20–92) 21.3 HC 28.0 (20–46) 8.2 FSMC MOTOR RRMS 28.8 (10–45) 10.6; HC 12.9 (10–23) 3.2 FSMC COGNITIVE RRMS 30.5 (10–48) 11.9; HC 15.0 (10–28) 5.6 | BDI > 16 excluded BDI-II: RRMS Mean/ Range/SD 7.2 (0–22) 6.0; HC 1.6 (0–11) 2.8 | T1W, T2W, FLAIR, fMRI | WM lesion load | N/A | fMRI: finger against thumb tapping | Participants with F had lower total intracranial volume compared to nF. Participants with F had lower recruitment of L-dorsal pre-motor cortex and L-dorsomedial PFC compared to nF. |
| Pravatà et al. [130] | F: FSMCcognitive≥22 (SD) HC 14.4 (±4.1); nF 14.3 (±3.8); F 33.6 (±4.8) | BDI ≥ 10 excluded | T1W, T2W, rs-fMRI | T2 lesion maps, Normalised brain, GM, and WM volumes | N/A | rs-fMRI: assessed right before, right after and 30 min after PASAT | No significant differences in structural measures between F and nF. Higher connectivity of the L/R-middle temporal gyrus and R-middle occipital gyrus with high FSMC scores. Fronto–temporal–occipital hyperconnectivity centred on the L- superior frontal gyrus in F-group. |
| Wu et al. [129] | MFIS Mean MFIS-5 (range): RRMS 11.2 (6–17); HC 0.7 (0–4) | N/A | T1W, T2W, rs-fMRI | Total WM lesion load | N/A | rs-fMRI | No significant structural correlations between F and nF RRMS patients. A high MFIS score was associated with increased FC between R-caudate and R-DLPFC. |
| Wu et al. [146] | Mean MFIS (range): acute RRMS 11.2 (6–16); relapsing RRMS 9.0 (2–15) | N/A | T2*W, T2W, T1W, rs-fMRI | N/A | N/A | rs-fMRI | No significant correlations. |
| Zhou et al. [147] | Mean MFIS-5 (range): 11.29 (6–17) 0.29 (0–1) | N/A | T2W, T1W, DTI, rs-fMRI | Total WM lesion load, GM, WM, and CSF volumes, BPF | DTI | rs-fMRI | Increased MD and RD of the tract linking the medial PFC and the L- inferior parietal lobule positively correlated with MFIS. No correlation between fatigue and conventional MRI or functional connectivity measures were reported. |
| Cruz Gomez et al. [148] | FSS mean (SD) nF:2.21 (0.96); F 5.6 (0.85) | Used our non-approved test | T1W, T2*W | ICV, GM, WM volumes | N/A | rs-fMRI | F patients showed extended GM and WM atrophy focused on areas related to the SMN. High FSS scores were associated with decreased rs-FC between the SMA and associative somatosensory cortex. Lower rs-FC in the premotor cortex in F patients. F patients exhibited GM atrophy in the paracentral gyrus (SMA), precentral gyrus (PMC), occipital lobe, precuneus, and posterior cingulate gyrus; F had reduced GM volume in the L-cerebellum; F patients showed WM alterations that extended into a larger number of brain regions in the frontal (including the motor areas and insula), temporal, occipital, and parietal lobes. F patients also showed WM atrophy around the thalamus, corpus callosum, and WM of cingulate gyrus (anterior, middle, and posterior parts); F patients showed WM atrophy in L-frontal areas that included the L-medial frontal gyrus of the SMA, L-superior frontal gyrus L-precuneus, brainstem, L-cerebellum. Higher FSS scores were associated with lower rs-FC between the SMA and PMC; Compared to F patients, nF also showed increased rs-FC in the L-precentral gyrus, in this case associated with the premotor cortex. |
| Bauer et al. [149] | FSMC total: Mean(Range)[SD] RRMS: 59.9 (20–92)[21.5]; HC: 28(19–46)[8.2]; FSMC motor: Mean(Range)[SD] RRMS: 29 (10–25)[10.4]; HC: 12.9(10–23)[3.2]; FSMC cognition: Mean(Range)[SD] RRMS: 30.9(10–48)[12.2]; HC: 15 (9–28)[5.6] | BDI–II Mean(Range) [SD] RRMS: 7.7(0–22)[6.4] HC: 1.5(0–11)[2.8] | T1W, T2W, FLAIR, DTI | Whole brain, GM, and WM volume; WML volume and number | DTI: anatomical connectivity mapping (ACM), FA, MD | N/A | F showed higher mean ACM values in the L relative to the R corticospinal tract-NAWM; MS showed a significant positive correlation between the left-right asymmetry of anatomical connectivity in the corticospinal tract and motor fatigue, but not cognitive fatigue. Left-right asymmetry in anatomical connectivity outside the corticospinal tract did not scale with individual motor fatigue. |
| Iancheva et al. [127] | F: MFIS ≥ 38 Mean(SD) CI: 12.43(12.1), CP: 12.17(15.91) | nD: BDI-II<10 BDI (mean ± SD): CI 4.30 ± 4.94; CP 2.67 ± 4.29 | T1W, fMRI | N/A | N/A | fMRI: numeric arithmetic task | High MFIS scores correlated with increased FC in supramarginal gyrus and premotor cortex, and decreased activation in posterior cingulate cortex. |

*(Continued)*

**Table 1.** (Continued)

| Authors | Fatigue assessment | Depression assessment | MRI sequence | Structural MRI | Diffusion MRI | Functional MRI | Major findings |
|---|---|---|---|---|---|---|---|
| *Gilio et al. [150]* | *MFIS(median[IQR]): 24 [6.75–34]* | *BDI-II(median[IQR]): 7 [2.25–11.75]* | *T1W, FLAIR, PD, FLAIR, Gd+ T1W* | *cortical thickness and T2 lesion load* | *NA* | *NA* | *T2 lesion load showed a positive correlation with MFIS scores.* |
| *Khedr et al. [151]* | *FSS: no data F n = 31; non-F n = 12; 31 patients (72.1%) had fatigue* | *HDRS: no data* | *PD, FLAIR,T1W, T2W, Gd+T1W* | *Total brain, cerebral and cerebellar GM and WM volume; hippocampus, thalamus, caudate, putamen volumes.* | *NA* | *NA* | *Total brain, cerebral grey matter, and thalamic volumes all had negative correlations with F. Thalamic and brainstem atrophy accounted for 50.7% variance in F scores.* |
| *Ruiz-Rizzo et al. [152]* | *MFIS(mean(SD)):35.8 (20.5)* | *HADS-D(mean(SD)): 11.1(7. 6)* | *T1W, T2*W, FLAIR, BOLD fMRI* | *Total WL volume* | *NA* | *Brain networks: 2 sensorimotor (lateral and central); 1 basal-ganglia; 2 default-mode (anterior and posterior); and 2 lateralized fronto-parietal (L and R)* | *Higher F was associated with lower FC of the precentral gyrus in the sensorimotor network, the precuneus in the posterior DMN and the superior frontal gyrus in the left fronto-parietal network. Associations between F and the sensorimotor network's global FC.* |
| *Svolgaard et al. [153]* | *FSMC(mean(range)SD): RRMS: 59.3 (20–92) 21.3 HC 28.0 (20–46) 8.2; FSMCmotor: RRMS: 28.8 (10–45) 10.6; HC: 12.9 (10–23) 3.2; FSMCcognitive: RRMS: 30.5 (10–48) 11.9; HC: 15.0 (10–28) 5.6* | *BDI-II(mean(range) SD): RRMS: 7.2 (0–22) 6.0; HC: 1.6 (0–11) 2.8* | *T1W, T2W, FLAIR, fMRI* | *lesion load and overall brain atrophy* | *NA* | *whole-brain fMRI* | *The more patients increased task-related activity in left dorsal premotor cortex after the fatiguing task, the less they experienced motor fatigue during daily life.* |
| Alshehri et al. [154] | *MFIS(mean±SEM): HC: 12±2.5; RRMS: 36 ±3.03; Physical F: HC: 5±0.93; RRMS: 18 ±1.50; Cognitive F: HC: 7±1.7; RRMS: 18 ±1.65* | NA | T1W, FLAIR, DTI | Total brain WM and WML volumes | FA, MD, RD, and AD | NA | No correlation between the fatigue domains and DTI metrics in total brain WM and WML volumes was observed |
| Tijhuis et al. [155] | *Checklist of Individual Strength (CIS-20r) Baseline RRMS: 74.36 (29.33); HC: 46.72 (17.06); Follow-up RRMS: 69.91 (27.01)a HC: 45.11 (19.84)* | *HADS-D(median (range)): Baseline: RRMS: 3(0–14); HC: 1(0–6); Follow up: RRMS: 3(0–9); HC: 1(0–12)* | *T1W, FLAIR, fMRI* | *WML, WM and GM volumes, subcortical segmentation* | *NA* | *Global static FC (sFC) and dynamic FC; regional connectivity basal ganglia and DMN (medial prefrontal, posterior cingulate and precuneal cortices)* | *Less dynamic connectivity between the basal ganglia and the cortex is associated with greater F.* |

Studies with positive findings are shown in italics.

AD: Axial Diffusivity, BDI: Beck Depression Index, BOLD: Blood-Oxygen Level Dependent, BPF: Brain Parenchymal Fraction, CES-D: Centre for Epidemiological Studies–Depression, CI: cognitively impaired, CIS: Clinically isolated syndrome, CNS: Central Nervous System, CP: cognitively preserved, DLPFC: dorsolateral prefrontal cortex, DMN: default mode network, D/nD: [not]depression/depressed, DSM-V: Diagnostic and Statistical Manual V semi-structured interview, DTI: Diffusion Tensor Imaging, EDSS: Expanded Disability Status Scale, F/nF: [not]fatigue/fatigued, FA: Fractional Anisotropy, FFG: fusiform gyrus, FIS: Fatigue Impact Scale, FLAIR: Fluid-attenuated Inversion Recovery, fMRI: functional Magnetic Resonance Imaging, FSMC: Fatigue Scale for Motor and Cognitive Functions, FSPGR: fast spoiled gradient echo, FSS: Fatigue Severity Scale, GD+: Gadolinium Enhancing, GM: Grey Matter, HARDI: high angular resolution diffusion imaging, HC: healthy controls, HDRS: Hamilton Depression Rating Scale, ICV: intracranial volume, L: left, LF/MF/HF: low/medium/high fatigue, MADRS: Montgomery-Asberg Depression Rating Scale, MD: Mean Diffusivity, MFIS: Modified Fatigue Impact Scale, N/A: not available, NAWM: normal appearing white matter, PASAT: Paced Auditory Serial Addition Test, (DL/M)PFC: (dorsolateral/medial) prefrontal cortex, PMC: premotor cortex, PPMS: Primary progressive MS, R: right, RD: Radial Diffusivity, ROI: region of interest, RRMS: Relapsing-remitting multiple sclerosis, rs-fMRI: Resting-state Functional Magnetic Resonance Imaging, SII: secondary sensorimotor cortex, SD: Standard Deviation, SEM: standard error of the measurements, SMA: supplementary motor area, SPMS: Secondary progressive MS, SSRI: Selective serotonin reuptake inhibitors, T2LV: T2 lesion volume, UF: uncinate fasciculus, WM: White matter.

## 3.2 Quality assessment

For quality assessment of cross-sectional studies, 28/52 studies fulfilled all criteria except for sample size justification and 46/52 studies fulfilled more than 80% of the criteria (S3 and S5 Tables). It should be noted that none of the assessed studies justified their sample sizes by *ad hoc* statistical power (Selection bias), therefore, not a single study was awarded full points. Out of 8 longitudinal studies, 7 fulfilled 70% or more criteria of the IHE checklist, and the lowest score was 50% (S4 and S6 Tables). The difference in average scores between cross-sectional and longitudinal studies should be attributed to different scales used.

## 3.3 Depression

**3.3.1 Conventional MRI measures.** Seventeen studies were identified that investigated associations between structural brain measures and depression (Table 1) [29, 50, 100–106, 136–139, 141, 142, 144, 145]. 10/17 studies did not find any associations (Table 1) [102, 103,

106, 122, 136–138, 142, 144, 145] and 7/17 reported significant associations (Tables 1 and 2) between structural measures and depression severity [29, 50, 100, 101, 104, 139, 141]. Of note, seven of these 17 studies investigated WML measures [29, 100, 102, 103, 105, 106, 145], but only three observed associations between depression and lesion load [29, 100, 141]. Kopchak and Odintsova observed that combined lesions in frontal lobe and corpus callosum were related to depressive scores [141].

Decreased volume of limbic structures was associated with high depression scores in 3/17 studies [50, 101, 104] (Fig 2). Additionally, changes in the frontal lobe were significantly associated with depression in 3/17 studies (Fig 2 and Table 2), specifically showing increased lesion load and reduced tissue volume in RRMS patients with high depression scores [50, 100, 141]. An association between lower volume of the cerebellar right Vermis Crus I and depression score was also observed [139], as well as an overall increase in T2 lesion burden in depressed pwRRMS [29].

**3.3.2 Structural connectivity.** Five studies were identified that assessed associations between structural connectivity measures and depression in pwRRMS, four of which used DTI [29, 102, 140, 142] and one used HARDI [143] (Tables 1 and 3), but only three found statistically significant relationships between structural connectivity and depression in pwRRMS [29, 140, 143].

An increased local path length between the right hippocampus and right amygdala, as well as 'shortest distance' (i.e., shortest distance between couples of brain nodes)—suggestive of reduced structural connectivity—between both the right hippocampus and the right amygdala and several regions, including the dorsolateral- and ventrolateral prefrontal cortex (DLPFC, VLPFC), and the orbitofrontal cortex correlated with high BDI scores [102] (Tables 1 and 3). The remaining studies observed a correlation between depression scores and decreased FA in the cingulum, uncinate fasciculus, and fornix [140], with decreased FA, and increased RD and MD in the right superior longitudinal fasciculus. In contrast, Rojas et al. did not detect any differences in global FA among pwRRMS with and without depression [29] and Golde et al. did not observe any correlation between DTI and depression measures [142].

**3.3.3 Functional connectivity.** Depression severity in relation to fMRI was examined in five studies (Tables 1 and 4) [103, 106, 138, 142, 145], of which four used rs-fMRI [106, 138, 142, 145] and one used task-based (emotional processing) fMRI [103]. Four studies reported significant findings [103, 106, 138, 145].

Firstly, Carotenuto et al. in their rs-fMRI study, reported altered functional connectivity between a wide number of brain regions: brainstem and hypothalamus; amygdala and cortical regions (including postcentral gyrus, supramarginal gyrus, cerebellum); cerebellum and amygdala, hippocampus, hypothalamus, locus coeruleus, nucleus accumbens, thalamus, ventral tegmental area in RRMS patients with high HDRS scores [106] (Table 1). Secondly, Riccelli et al. reported negative correlations between BDI and functional connectivity of the hippocampus with orbitofrontal cortex and DLPFC; the amygdala and DLPFC; and an association between reduced activity of the subgenual cingulate cortex and depression severity, in a task-based fMRI study [103] (Table 1). Furthermore, Jaeger et al. observed associations between altered functional connectivity in regions of the sensory motor cortex (precentral, postcentral gyri) and the superior ventral striatum and high BDI scores [138]. Lastly, Romanello et al. related depression severity to functional connectivity of the ventral attention network with the dorsal attention network and DMN [145].

## 3.4 Fatigue

**3.4.1 Conventional MRI measures.** Forty-eight studies were identified that investigated associations between structural brain abnormalities and fatigue in pwRRMS (Table 1) [35, 50,

**Table 2. Brain regions suggested to be involved in depression and/or fatigue symptomatology in pwRRMS, assessed using conventional MRI, in 17/56 publications with positive findings.**

| Brain region | Depression | Overlap | Fatigue |
|---|---|---|---|
| Brainstem | - | - | WM atrophy (Cruz Gomez et al. [148]); atrophy (Khedr et al. [151]) |
| Caudate nucleus | - | - | atrophy (Andreasen et al. [108]); GM and WM atrophy (Calabrese et al. [110]); atrophy (Damasceno et al. [113]); atrophy (Khedr et al. [151]) |
| Cerebellum | Vermis Crus I atrophy (Lazzarotto et al. [139]) | Vermis Crus I atrophy (depression)/ lobule right V atrophy (fatigue) (Lazzarotto et al. [139]); cortical lesion volume (Damasceno et al. [113]); WM atrophy (Cruz Gomez et al. [148]) | lobule right V atrophy (Lazzarotto et al. [139]); cortical lesion volume (Damasceno et al. [113]); WM atrophy (Cruz Gomez et al. [148]) |
| L-cingulum cingulate bundle | - | - | structural damage (Pardini et al. [117]) |
| Cingulate gyrus | - | - | WM atrophy (Cruz Gomez et al. [148]); atrophy (Andreasen et al. [108]) |
| Corona radiata | - | - | lesion location (Altermatt et al. [107]) |
| Corpus callosum | lesions (Kopchak and Odintsova [141]) | lesions (Kopchak and Odintsova [141]); atrophy (Yaldizli et al. [122]); WM atrophy (Cruz Gomez et al. [148]) | atrophy (Yaldizli et al. [122]); WM atrophy (Cruz Gomez et al. [148]) |
| Neocortical gray matter | atrophy (Rojas et al. [29]) | - | - |
| L- superior frontal gyrus | - | - | atrophy (Andreasen et al. [108]) |
| L- medial frontal gyrus | - | - | WM atrophy (Cruz Gomez et al. [148]) |
| R-middle frontal region | - | - | reduced cortical volume (Nygaard et al. [50]) |
| Superior frontal gyrus | - | - | atrophy (Andreasen et al. [108]); WM atrophy (Calabrese et al. [110]) |
| Frontal lobe | area and number of brain lesions (Benesova et al. [100]); cortical surface area and volume (Nygaard et al. [50]); lesions (Kopchak and Odintsova [141]) | - | - |
| Frontal pole | cortical surface area and volume (Nygaard et al. [50]) | - | - |
| Fronto-temporal region | lesions (Kopchak and Odintsova [141]) | - | - |
| Middle temporal fusiform gyrus | atrophy (Nygaard et al. [50]) | - | - |
| Hippocampus | atrophy (Gold et al. [101]) | - | - |
| L-anterior insula | - | - | atrophy (Andreasen et al. [108]) |
| Nucleus accumbens | - | - | atrophy (Damasceno et al. [113]) |
| Occipital lobe | - | - | WM and GM atrophy (Cruz Gomez et al. [148]) |
| Pallidum | - | - | atrophy (Yarraguntla et al. [123]) |
| Paracentral gyrus | - | - | GM atrophy (Cruz Gomez et al. [148]) |
| Parahippocampal gyrus | reduced cortical surface area (Nygaard et al. [50]) | - | - |

*(Continued)*

**Table 2.** (Continued)

| Brain region | Depression | Overlap | Fatigue |
|---|---|---|---|
| Inferior parietal gyrus | - | - | atrophy (Andreasen et al. [108]); cortical atrophy (Calabrese et al. [110]) |
| L-inferior parietal lobe | reduced cortical surface area and atrophy (Nygaard et al. [50]) | - | - |
| R-superior parietal lobe | - | - | atrophy (Andreasen et al. [108]) |
| Pars Orbitalis | reduced cortical surface area and atrophy (Nygaard et al. [50]) | - | - |
| Postcentral gyrus | reduced cortical surface area and atrophy (Nygaard et al. [50]) | reduced cortical surface area and atrophy (D)/reduced cortical volume (F) (Nygaard et al. [50]) | Reduced cortical volume (Nygaard et al. [50]) |
| Precentral gyrus | reduced cortical surface area and atrophy (Nygaard et al. [50]) | reduced cortical surface area and atrophy (D)/reduced cortical volume (F) (Nygaard et al. [50]); GM atrophy (Cruz Gomez et al. [148]) | GM atrophy (Cruz Gomez et al. [148]); reduced cortical volume (Nygaard et al. [50]) |
| L-precuneus | - | - | WM and GM atrophy (Cruz Gomez et al. [148]) |
| Putamen | - | - | atrophy (Calabrese et al. [110]) |
| R-middle temporal gyrus | - | - | atrophy (Andreasen et al. [108]) |
| R-superior temporal gyrus | atrophy (Nygaard et al. [50]) | atrophy (Nygaard et al. [50]) atrophy (Andreasen et al. [108]) | atrophy (Andreasen et al. [108]) |
| L-interior temporal lobe | atrophy (Nygaard et al. [50]) | - | - |
| Thalamus | atrophy (Štecková et al. [104]) | atrophy (Calabrese et al. [110]); increased volume (Damasceno et al. [113]); lesion load (Wilting et al. [121]); atrophy (Štecková et al. [104]); atrophy (Saberi et al. [131]); WM atrophy (Cruz Gomez et al. [148]); atrophy (Khedr et al. [151]) | atrophy (Calabrese et al. [110]); increased volume (Damasceno et al. [113]); lesion load (Wilting et al. [121]); atrophy (Saberi et al. [131]); WM atrophy (Cruz Gomez et al. [148]); atrophy (Khedr et al. [151]) |

pwRRMS = patients with relapsing-remitting multiple sclerosis, MRI = magnetic resonance imaging, WM = white matter, GM = gray matter, L = left, R = right, D = depression, F = fatigue

70, 107–126, 128–139, 142–145, 147–155]. 33/48 studies did not observe any associations (Table 1) [35, 70, 109, 111, 112, 114–116, 118, 120, 124–126, 128–130, 132–138, 142–145, 147, 149, 152–155] and 15/48 reported significant associations (Tables 1 and 2) between fatigue and structural brain changes. Of note, 28/48 studies investigated WMLs, but only three studies found significant associations between fatigue and WMLs [107, 113, 150], and one observed a link between motor fatigue and cortical lesions [121].

Five studies out of 48 linked fatigue in pwRRMS to thalamic atrophy [110, 113, 131, 148, 151] and one to lesion load in the thalamus [121]. Moreover, 4/48 studies associated fatigue with cerebellar atrophy [113, 139, 148, 151] and 4/48 with decreased volume of caudate nucleus [108, 110, 113, 151]. Additionally, fatigue was associated with the atrophy in basal ganglia structures [108, 110, 113], inferior parietal gyrus [108, 110] and corpus callosum [122, 148] (Fig 2 and Table 2). Furthermore, the remaining studies observed correlations between fatigue scores and several regions in the parietal, frontal, insular and temporal lobes, as well as the cingulate gyrus [108]; the occipital lobe, brainstem, and cingulate gyrus [148] and a weak correlation was detected between motor fatigue and WML volumes [121]. In contrast however, four studies reported an absence of associations between thalamic atrophy and fatigue scores [116, 120, 124, 129]. Similarly, the absence of association was reported between fatigue and basal ganglia volume [112], the limbic system [116], and amygdala volume [144].

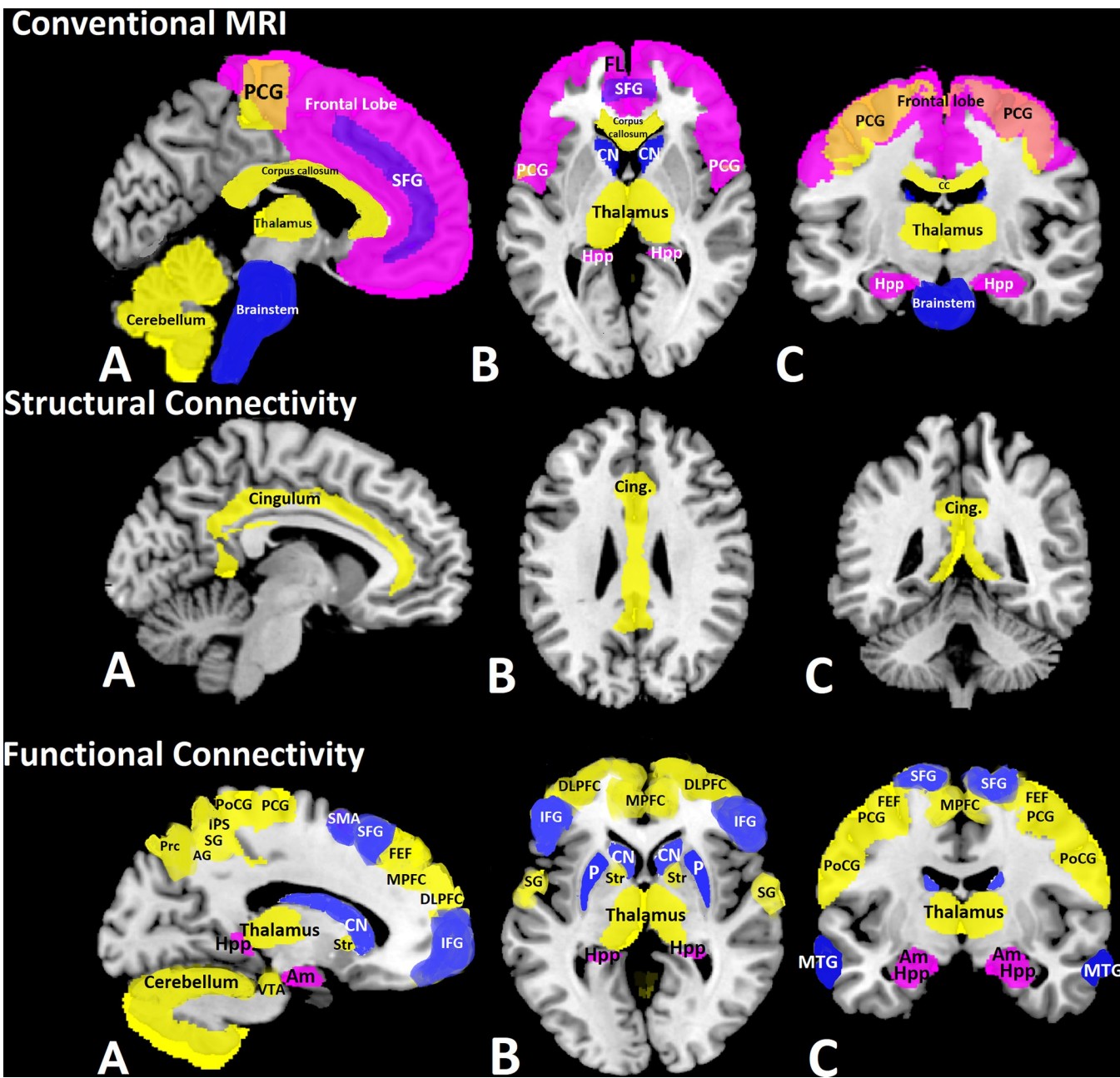

**Fig 2.** Sagittal (A), axial (B), and coronal (C) view of brain regions suggested to be involved in depression (magenta), fatigue (blue) or both* (yellow) in >1 study, using conventional MRI, structural and functional connectivity. Brain regions were extracted from brain atlases available in FSL [156] and superimposed on a template T1w image, available in MRIcron [157]. Results from included publications were compiled and summarised in this figure by the authors of this study, using MRIcron [157]. *AG: angular gyrus (as a region of default mode network), Am: amygdala, CC: corpus callosum, Cing: cingulum, CN: caudate nucleus, DLPFC: dorsolateral prefrontal cortex, FEF: frontal eye field (as a region of dorsal attention network), FL: frontal lobe, Hpp: hippocampus, IFG: inferior frontal area, IPS: intraparietal sulcus (as a region of dorsal attention network), MPFC: medial prefrontal cortex (as a region of default mode network), MTG: middle temporal gyrus, P: putamen, PCG: precentral gyrus, PoCG: postcentral gyrus, Prc: precuneus (as a region of default mode network), SFG: superior frontal gyrus, SG: supramarginal gyrus, SMA: supplementary motor area, STG: superior temporal gyrus, Str: superior ventral striatum, VTA: ventral tegmental area.* *Overlapping brain regions between symptoms, in at least 1 study for each symptom.

**Table 3. Brain regions suggested to be involved in depression and/or fatigue symptomatology in pwRRMS structural connectivity measures, in 10/18 publications with positive findings.**

| Brain region/network | Depression | Overlap | Fatigue |
|---|---|---|---|
| R-amygdala and orbitofrontal cortex (e.g., rectus gyrus) | increased shortest distance (Nigro et al. [102]) | - | - |
| R-amygdala and DLPFC | increased shortest distance (Nigro et al. [102]) | - | - |
| R-amygdala and ventrolateral PFC (i.e., inferior frontal gyrus) | increased shortest distance (Nigro et al. [102]) | - | - |
| Anterior thalamic radiation | - | - | reduced FA (Pardini et al. [118]) |
| Basal Ganglia (Caudate Nucleus, Pallidus, Putamen) | - | - | reduced FA and increased MD (Wilting et al. [121]) |
| Cingulum | reduced FA (Hassan et al. [140]) | reduced FA (Hassan et al. [140]); reduced FA (Pardini et al. [117]); reduced FA (Pardini et al. [118]) | reduced FA (Pardini et al. [117]); reduced FA (Pardini et al. [118]) |
| Anterior/medial cingulate cortices | - | - | reduced FA (Pardini et al. [117]) |
| Corticospinal tract | - | - | higher connectivity in the L vs R hemisphere (Bauer et al. [149]) |
| Forceps minor | - | - | reduced FA (Pardini et al. [118]) |
| Fornix | reduced FA (Hassan et al. [140]) | - | - |
| R-hippocampus and rectus gyrus | increased shortest distance (Nigro et al. [102]) | - | - |
| R-hippocampus and DLPFC | increased shortest distance (Nigro et al. [102]) | - | - |
| R-hippocampus and ventrolateral PFC (i.e., inferior frontal gyrus) | increased shortest distance (Nigro et al. [102]) | - | - |
| Inferior fronto-occipital fasciculus | - | - | reduced FA (Pardini et al. [118]) |
| L-internal capsule | - | - | reduced FA (Pardini et al. [118]) |
| Medial PFC and L-inferior parietal lobule | - | - | increased MD and RD (Zhou et al. [147]) |
| R-temporal lobe | - | - | reduced FA (Yarraguntla et al. [123]) |
| Thalamocortical tracts | - | - | increased MD, RD, and AD (Zhou et al. [125]) |
| Thalamus | - | - | reduced FA and increased MD (Wilting et al. [121]) |
| Uncinate fasciculus | reduced FA (Hassan et al. [140]) | - | - |
| R-superior longitudinal fasciculus | reduced FA, increased RD, and MD (Beaudoin et al. [143]) | - | - |

pwRRMS = patients with relapsing-remitting multiple sclerosis, R = right, L = left, (DL)PFC = (dorsolateral)prefrontal cortex, FA = fractional anisotropy, MD = mean diffusivity, RD = radial diffusivity

**3.4.2 Structural connectivity.** Fifteen studies were identified that evaluated the relationship between fatigue and dMRI measures, all of which used DTI and one used HARDI [143]. Seven out of fifteen studies did not report any significant findings (Table 1) [70, 109, 112, 114, 142, 143, 154] and 8/15 found significant associations (Tables 1 and 3).

Two studies out of fifteen observed negative correlations between cingulum FA and fatigue scores in pwRRMS (Fig 2 and Table 3) [117, 118]. In addition, the remaining studies reported a correlation of fatigue in RRMS patients with lower FA in the inferior occipitofrontal fasciculus, internal capsule, anterior thalamic radiation, and forceps minor [118]; a lower number of connectivity streamlines in the corticospinal tract [149], and reduced FA and increased MD values of the thalamus and basal ganglia [121]. Moreover, fatigue correlated with lower FA in the right temporal cortex, and higher MD, RD, and AD in the thalamocortical tracts [123];

**Table 4. Brain regions suggested to be involved in depression and/or fatigue in pwRRMS using functional connectivity measures, in 19/22 publications with positive findings.**

| Brain region/ network | Depression | Overlap | Fatigue |
|---|---|---|---|
| Amygdala | increased rs-FC with L-supramarginal gyrus, L-postcentral gyrus, cerebellum (Carotenuto et al. [106]); reduced task-based FC with L-DLPFC (Riccelli et al. [103]) | - | - |
| Associative somatosensory cortex | - | - | reduced rs-FC with supplementary motor area (Cruz Gomez et al. [148]) |
| Attention networks | - | - | reduced rs-FC with superior ventral striatum (Jaeger et al. [138]) |
| Dorsal attention network | increased rs-FC with ventral attention network (Romanello et al. [145]) | increased rs-FC (Romanello et al. [145]); reduced activity (Huang et al. [115]) | reduced activity (Huang et al. [115]) |
| Ventral attention network | increased rs-FC with dorsal attention network, default mode network (Romanello et al. [145]) | - | - |
| Basal ganglia | - | - | reduced dynamic FC with default mode network (Tijhuis et al. [155]); reduced rs-FC with medial PFC, posterior cingulate cortex, precuneus (Finke et al. [114]); increased activation (Specogna et al. [128]); reduced activation (Rocca et al. [133]); reduced rs-FC with frontoparietal network (Romanello et al. [145]) |
| Brainstem | reduced rs-FC with L-hypothalamus (Carotenuto et al. [106]) | - | - |
| Caudate nucleus | - | - | reduced rs-FC with motor cortex, precentral gyrus (Finke et al. [114]); reduced activation (Rocca et al. [35]); increased rs-FC with R-DLPFC (Wu et al. [129]); reduced rs-FC with sensorimotor cortex (supplementary motor area, precentral, postcentral gyrus), middle frontal gyrus, parietal lobule, precuneus, superior frontal cortex, L-superior ventral striatum (Jaeger et al. [138]); increased rs-FC with superior frontal gyrus (Pravatà et al. [130]) |
| Cerebellum | reduced rs-FC with L-hippocampus, L-nucleus accumbens, locus coeruleus, R-hypothalamus, increased rs-FC with R-amygdala, R-thalamus, ventral tegmental area (Carotenuto et al. [106]) | reduced/increased rs-FC (Carotenuto et al. [106]); increased activation (Rocca et al. [133]); reduced activation (Specogna et al. [128]); reduced activation (Filippi et al. [126]); increased task-related activity (Svolgaard et al. [119]) | increased activation task-related activation of posterior lobe (Rocca et al. [133]); reduced activation (Specogna et al. [128]); reduced activation (Filippi et al. [126]); increased task-related activity of R-upper cerebellar lobule VI (Svolgaard et al. [119]) |
| Posterior cingulate cortex | - | - | reduced global FC (Ruiz-Rizzo et al. [152]); reduced activation (Iancheva et al. [127]); reduced rs-FC basal ganglia, pallidum, putamen (Finke et al. [114]) |
| R-subgenual cingulate gyrus | reduced activity (Riccelli et al. [103]) | - | - |
| Cingulate motor area | - | - | reduced activation (Rocca et al. [133]); increased activation (Filippi et al. [126]) |

*(Continued)*

**Table 4.** (Continued)

| Brain region/ network | Depression | Overlap | Fatigue |
|---|---|---|---|
| Default mode network | increased rs-FC with ventral attention network (Romanello et al. [145]) | increased rs-FC (Romanello et al. [145]); reduced activity (Huang et al. [115]); reduced dynamic FC (Tijhuis et al. [155]) | reduced activity (Huang et al. [115]); reduced dynamic FC with basal ganglia (Tijhuis et al. [155]) |
| L- Inferior frontal gyrus | - | - | reduced task-based activation (Rocca et al. [133]); reduced activation (Filippi et al. [126]) |
| Middle frontal gyrus | - | - | increased activation (Specogna et al. [128]); increased activity (Rocca et al. [35]); reduced rs-FC (Finke et al. [114]); reduced activation (Filippi et al. [126]); reduced rs-FC with superior ventral striatum, caudate nucleus (Jaeger et al. [138]) |
| Superior frontal gyrus | - | - | reduced activation (Rocca et al. [35]); increased rs-FC with middle occipital gyri, temporal area, frontal area, L-caudate nucleus, reduced connectivity with L-anterior thalamus (Pravatà et al. [130]); reduced global FC (Ruiz-Rizzo et al.[152]); reduced rs-FC with caudate nucleus (Jaeger et al. [138]) |
| Frontoparietal network | - | - | reduced rs-FC with basal ganglia (Romanello et al. [145]) |
| Fusiform gyrus | - | - | increased rs-FC with medial prefrontal cortex, reduced rs-FC with R-lateral prefrontal cortex (Golde et al. [142]) |
| Hippocampus | reduced task-based FC with L-DLPFC, orbitofrontal cortex (Riccelli et al. [103]); reduced rs-FC with cerebellum (Carotenuto et al. [106]) | - | - |
| Hypothalamus | reduced rs-FC with brainstem, cerebellum (Carotenuto et al. [106]) | - | - |
| Locus coeruleus | reduced rs-FC with cerebellum (Carotenuto et al. [106]) | - | - |
| Motor cortex | - | - | reduced rs-FC with caudate nucleus (Finke et al. [114]) |
| Primary motor cortex | - | - | reduced rs-FC with precentral gyrus, supplementary motor area (Cruz Gomez et al. [148]) |
| Nucleus accumbens | reduced rs-FC with cerebellum (Carotenuto et al. [106]) | - | - |
| Nucleus basalis (Ch4) | increased rs-FC with R-angular gyrus (Carotenuto et al. [106]) | - | - |
| Middle occipital gyri | - | - | increased rs-FC with superior frontal gyrus (Pravatà et al. [130]) |
| orbitofrontal cortex | reduced task-based FC with hippocampus (Riccelli et al. [103]) | - | - |
| Pallidum | - | - | reduced rs-FC with precuneus, posterior cingulate cortex, dorsal and ventral medial prefrontal cortex (Finke et al. [114]) |
| R-parahippocampal gyrus | - | - | increased rs-FC (Huang et al. [115]) |
| Parietal lobule | - | - | reduced rs-FC with caudate nucleus (Jaeger et al. [138]) |

(*Continued*)

**Table 4.** (Continued)

| Brain region/ network | Depression | Overlap | Fatigue |
|---|---|---|---|
| Parietal operculum | - | - | increased rs-FC with dorsolateral prefrontal cortex, superior ventral striatum (Jaeger et al. [138]) |
| Postcentral gyrus | increased rs-FC with amygdala (Carotenuto et al. [106]); reduced rs-FC with superior ventral striatum (Jaeger et al. [138]) | increased rs-FC (Carotenuto et al. [106]); reduced rs-FC [depression], increased rs-FC [fatigue] (Jaeger et al. [138]); reduced activation of L- (Rocca et al. [35]); reduced task-based activation of L-(Rocca et al. [133]) | reduced activation of L- (Rocca et al. [35]); reduced task-based activation of L-(Rocca et al. [133]); increased rs-FC with dorsolateral prefrontal cortex, reduced rs-FC with superior ventral striatum, caudate nucleus (Jaeger et al. [138]) |
| Precentral gyrus | reduced rs-FC with superior ventral striatum (Jaeger et al. [138]) | reduced rs-FC (Jaeger et al. [138]); reduced rs-FC (Cruz Gomez et al. [148]); reduced global FC (Ruiz-Rizzo et al. [152]); reduced task-based activation (Rocca et al. 2009 [133]); reduced rs-FC (Finke et al. [114]) | reduced rs-FC with primary motor cortex (Cruz Gomez et al. [148]); reduced global FC (Ruiz-Rizzo et al. [152]); reduced task-based activation (Rocca et al. 2009 [133]); reduced rs-FC with caudate nucleus (Finke et al. [114]); reduced rs-FC with caudate nucleus, superior ventral striatum, increased rs-FC with dorsolateral prefrontal cortex (Jaeger et al. [138]) |
| Precuneus | - | - | increased activation (Rocca et al. [133]); reduced global FC (Ruiz-Rizzo et al. [152]); reduced activation (Filippi et al. [126]); reduced rs-FC with basal ganglia, pallidum, putamen (Finke et al. [114]); reduced rs-FC with caudate nucleus (Jaeger et al. [138]) |
| Dorsolateral prefrontal cortex | reduced task-based FC with amygdala, hippocampus (Riccelli et al. [103]) | reduced task-based FC (Riccelli et al. [103]); increased activation (Specogna et al. [128]); increased rs-FC (Jaeger et al. [138]); lack of increased task-based activation (Svolgaard et al. [119]); increased rs-FC (Wu et al. [129]) | increased activation (Specogna et al. [128]); increased rs-FC with parietal operculum, supramarginal gyrus, postcentral gyrus, precentral gyrus (Jaeger et al. [138]); lack of increased task-based activation (Svolgaard et al. [119]); increased rs-FC with caudate nucleus (Wu et al. [129]) |
| R-lateral prefrontal cortex | - | - | reduced rs-FC with fusiform gyrus (Golde et al. [142]) |
| Medial prefrontal cortex | - | - | reduced rs-FC with basal ganglia, pallidum, putamen (Finke et al. [114]); increased rs-FC with fusiform gyrus (Golde et al. [142]) |
| Premotor area | - | - | increased activation (Specogna et al. [128]); increased activation (Iancheva et al. [127]) |
| L-dorsal premotor cortex | - | - | reduced task-related activation (Svolgaard et al. [153]) |
| L-pre-supplementary motor area | - | - | reduced activation (Rocca et al. [35]) |
| L-primary sensorimotor cortex | - | - | increased task-related activity (Rocca et al. [133]) |
| Putamen | - | - | reduced rs-FC with dorsal/ventral medial prefrontal cortex, posterior cingulate cortex, precuneus (Finke et al. [114]); reduced activation (Rocca et al. [35]); increased activation (Specogna et al. [128]) |

(*Continued*)

**Table 4.** (Continued)

| Brain region/ network | Depression | Overlap | Fatigue |
|---|---|---|---|
| Reward networks | - | - | reduced rs-FC with superior ventral striatum (Jaeger et al. [138]) |
| Rolandic operculum | - | - | reduced activation (Filippi et al. [126]) |
| L-secondary sensorimotor cortex | - | - | increased task-based activation (Rocca et al. [133]) |
| Superior ventral striatum | reduced rs-FC with precentral, postcentral gyrus (Jaeger et al. [138]) | reduced rs-FC (Jaeger et al. [138]) | reduced rs-FC with attention networks, reward networks, sensorimotor cortex (supplementary motor area, precentral, postcentral gyrus), parietal lobule, middle frontal gyrus, inferior temporal gyrus (Jaeger et al. [138]) |
| Supplementary motor area | - | - | reduced activation (Rocca et al. [35]); reduced rs-FC with associative somatosensory cortex, primary motor cortex (Cruz Gomez et al. [148]); reduced rs-FC with caudate nucleus, superior ventral striatum (Jaeger et al. [138]) |
| Supramarginal gyrus | increased rs-FC with amygdala (Carotenuto et al. [106]) | increased rs-FC (Carotenuto et al. [106]); increased activation (Iancheva et al. [127]); increased rs-FC (Jaeger et al. [138]) | increased activation (Iancheva et al. [127]); increased rs-FC with dorsolateral prefrontal cortex (Jaeger et al. [138]) |
| Inferior temporal gyrus | - | - | reduced rs-FC with superior ventral striatum (Jaeger et al. [138]) |
| Middle temporal gyrus | - | - | increased activity in the R-, reduced activity in the L- (Rocca et al. [35]); increased activation (Rocca et al. [133]) |
| R-superior temporal gyrus | - | - | increased connectivity coefficient (Huang et al. [115]) |
| Thalamus | increased rs-FC with cerebellum (Carotenuto et al. [106]) | increased rs-FC (Carotenuto et al. [106]); reduced activation (Filippi et al. [126]); reduced activation (Rocca et al. [133]); reduced FC (Pravatà et al. [130]) | reduced activation (Filippi et al. [126]); reduced activation (Rocca et al. [133]); reduced rs-FC with superior frontal gyrus (Pravatà et al. [130]) |
| Ventral tegmental area | increased rs-FC with cerebellum (Carotenuto et al. [106]) | increased rs-FC (Carotenuto et al. [106]); reduced rs-FC (Jaeger et al. [138]) | reduced rs-FC with superior ventral striatum (Jaeger et al. [138]) |

pwRRMS = patients with relapsing-remitting multiple sclerosis, (rs-)FC = resting state functional connectivity, DLPFC = dorsolateral prefrontal cortex, L = left,, R = right

and increased MD and RD of the WM tract connecting two DMN regions (i.e., medial pre-frontal cortex and inferior parietal gyrus–the WM tract was not further specified) [147]; and increased 'shortest distance' between both the right hippocampus and right amygdala and a series of regions including the dorsolateral and ventrolateral prefrontal cortex, orbitofrontal cortex, sensory-motor cortices and SMA [102].

**3.4.3 Functional connectivity.** Seven out of twenty studies looking at fatigue and fMRI used a task-based approach [35, 119, 126–128, 133, 153], and 13/20 used rs-fMRI [114, 115, 125, 129, 130, 138, 142, 145–148, 152, 155] (Table 1). Only 3/20 studies [125, 146, 147] did not observe functional changes in fatigued RRMS patients, while seventeen out of twenty studies reported associations with fatigue for one or more regions (Fig 2 and Table 4).

Most DMN regions displayed altered resting-state connectivity in association with high fatigue scores [115, 155], with the precuneus [114, 126, 133, 138, 152], medial prefrontal cortex [114, 142], posterior cingulate cortex [114, 127, 152] observed in more than one study (Table 2). Moreover, fatigue was linked with altered dynamic, resting-state functional connectivity and activation of the basal ganglia [114, 128, 133, 145, 155], including putamen [35, 114, 128], pallidum [114], superior ventral striatum [138]. Additionally, altered functional connectivity in the regions of the frontal [114, 138] (middle [35, 114, 126, 128, 138] and superior [35, 130, 138, 152], L-inferior [126, 133] gyri, dorsolateral prefrontal cortex [119, 128, 129, 138], L-dorsal premotor cortex [153]), occipito-temporal [142] (middle temporal gyrus [35, 133], R-superior temporal gyrus and R-parahippocampal gyrus [115], and inferior temporal gyrus [138], and middle occipital gyri [130]), and parietal [138] (postcentral gyrus [35, 133, 138], supramarginal gyrus [127, 138], associative somatosensory cortex [148], parietal operculum [138]) lobes were associated with fatigue in pwRRMS. Furthermore, changes in functional connectivity and of the motor area, including precentral gyrus [114, 133, 138, 148, 152], supplementary motor area [35, 138, 148], premotor area [127, 128], cingulate motor area [126, 133], motor cortex [114], primary motor cortex [148], L-pre-supplementary motor area [35], L-primary and L-secondary sensorimotor cortex [133] were associated with fatigue. Additionally, associations between functional changes of the caudate nucleus and fatigue were observed in four rs-fMRI studies [114, 129, 130, 138] and one task-based fMRI study assessing motor processing through finger-tapping and the nine-hole peg test [35] (Table 2). Moreover, changes in cerebellum activation [119, 126, 128, 133] was associated with fatigue in four task-based fMRI studies. Lastly, decreased activation and rs-FC in the thalamus [126, 130, 133] and attention networks [115, 138] were linked to fatigue scores in more than one study per symptom.

## 3.5 Fatigue and depression: Overlap

**3.5.1 Studies investigating both depression and fatigue together.**   Eight studies assessed both fatigue and depression [136–139, 142–145]. All but one [143] examined the associations between depression or fatigue and structural MR measures, but only one paper observed overlapping changes. Specifically, Lazzarotto et al. reported significant correlations between BDI scores and lower volume of the right cerebellar vermis crus I, and between FSS score and reduced volume of cerebellar lobule right V, but other than cerebellum involvement for both, no other overlapping brain areas were found [139]. The six remaining studies found no significant correlations between conventional MRI and depression or fatigue scores [136–138, 142, 144, 145]. Likewise, two studies that used DTI reported no association between diffusion measures and fatigue or depression [142, 143]. Out of two studies studying functional connectivity, only Jaeger et al. reported two overlapping brain areas using rs-fMRI [138]. Specifically, they observed negative correlations of both BDI and FSS scores with functional connectivity of the ventral striatum and post-central gyrus [138]. Golde et al., on the other hand, found no overlap between the two symptoms and rs-fMRI measures [142].

Given the small number of studies studying depression and fatigue together, and the lack of overlap, the five studies were included in the total counts/summaries of studies investigating depression and fatigue separately.

**3.5.2 Studies focusing on either depression or fatigue alone.**   Six out of eleven publications studying *only* depression in relation to MRI measures did not include fatigue assessments [100, 101, 104, 106, 140, 141], and no studies excluded individuals with high fatigue scores. The remaining 5/11 studies either controlled for fatigue status [29, 50, 105] or included fatigue as a covariate or a clinical symptom of no interest [102, 103]. Of the 41 publications reporting results of MRI measures in relation to fatigue *only*, 29/41 included depression assessments [35,

108–111, 113, 114, 118–123, 126–128, 130–135, 150–155], with 14/29 excluding participants with high depression scores [35, 108, 109, 114, 118, 121–123, 126, 128, 130, 131, 133, 135] and 10/29 controlling for depression status [110, 111, 113, 127, 132, 134, 149, 150, 154, 155], or both (5/29) [119, 120, 151–153].

**3.5.3 Overlapping brain regions.** For conventional MRI, several brain structures suggested to be associated with depression severity were also observed to be involved in fatigue in pwRRMS. Specifically, thalamic [104, 110, 113, 121, 131, 148, 151], cerebellar [113, 139, 148], corpus callosum [122, 141, 148], right superior temporal region [50, 108] and precentral gyrus [50, 148] volumes were negatively correlated with depression and fatigue scores in at least one study per symptom (Fig 2 and Table 2). For structural connectivity, overlap between associations reported for dMRI measures and fatigue or depression was observed in the cingulum (Fig 2 and Table 3). Meanwhile, functional connectivity changes of the thalamus [106, 126, 130, 133], cerebellum [106, 119, 126, 128, 133], and DLPFC [103, 119, 128, 129, 138] were observed in association with fatigue or depression in at least one study per symptom (Fig 2 and Table 4). Additionally, the postcentral- [35, 106, 133, 138] and precentral gyrus [114, 133, 138, 148, 152] of the SMN, supramarginal gyrus [106, 127, 138], DMN [115, 145, 155], dorsal attention network [115, 145], ventral tegmental area [106, 138] and superior ventral striatum [138] showed altered functional connectivity associated with depression and fatigue scores in at least one study per symptom (Table 1).

## 4. Discussion

This study systematically examined the literature for conventional MRI, structural and functional brain connectivity features associated with fatigue and depression in individuals with RRMS. Brain connectivity changes underlying fatigue have been observed in the cortico-thalamic-basal ganglial networks, while abnormal connectivity in the cortico-limbic networks was associated with depression. Some overlapping changes in depression and fatigue were observed for structural connectivity of the cingulum, and functional connectivity of the cerebellum, thalamus, frontal lobe, supramarginal gyrus, ventral tegmental area, superior ventral striatum, DMN, attention networks, and pre/post-central gyri. Overall, the literature reported mixed results, with half of the studies observing no significant associations and a limited number of studies investigating brain connectivity changes underlying depression in pwRRMS.

### 4.1 Brain connectivity changes underlying depression in pwRRMS

**4.1.1 Cortico-limbic network.** Depression in pwRRMS was associated with areas of the limbic system, especially the hippocampus and amygdala in five included studies. Nigro reported structural connectivity changes between the hippocampus, amygdala, and frontal areas in RRMS patients with depression [102]. Functional connectivity changes of the amygdala and hippocampus were also observed [103, 106] as was hippocampal atrophy [101, 104]. Their involvement in depression is perhaps unsurprising as both regions are associated with emotion-related functions [158]. The limbic system in general is thought to be responsible for emotional responses, long-term memory, fear conditioning, sleep, motivation, and social cognition [159], many of which are involved in depression. The hippocampus specifically is a part of the cholinergic system—involved in arousal, attention, cognition, and memory—and relates to emotion-regulating brain regions [160]. The amygdala is linked to emotion regulation and memory, as well as fear conditioning [161]. Previous literature supports the role of hippocampal and amygdala involvement in major depression disorder (MDD). The hippocampus, in particular, plays a key role in depression [161], with ample studies observing hippocampal atrophy and functional changes in MDD [162–166]. It has also been previously suggested that

neuroinflammation in the hippocampus contributes to development of depression in mixed subtype MS [167]. Studies have also shown altered amygdala functional connectivity in depression in MS of various types [168, 169] as well as abnormal functional connectivity between the amygdala and other brain regions in people with MDD [170].

*4.1.1.1 Fronto-limbic network*: *PFC.* Disrupted connectivity between limbic structures and the frontal lobe may be underlying depressive symptomatology in pwRRMS, according to five included studies. RRMS patients with high depression scores showed structural connectivity changes in several regions of the fronto-limbic network, i.e., between the hippocampus or amygdala and the PFC, which are all involved in emotional behaviour, cognition, and motor control [102, 106]. This is in line with previous research showing that abnormal structural connectivity of the fronto-limbic network may be evident in MDD [171, 172]. Furthermore, functional connectivity between the DLPFC and limbic structures was also linked to depression in pwRRMS [103]. The DLPFC controls working memory, goal-directed action, abstract reasoning and attention, and impairments of these functions may contribute to depression [173].

*4.1.1.2 Orbitofrontal cortex and cingulate cortex.* Additionally, functional connectivity changes between the orbitofrontal cortex and hippocampus [103], as well as orbital frontal atrophy [50], were also related to depression in pwRRMS. As the orbitofrontal cortex has a key role in emotion and decision-making, as well as reward circuits [174], its association with MDD is not surprising [175]. Moreover, functional connectivity changes of the subgenual anterior cingulate cortex (ACC) [103], as well as the cholinergic network (e.g., nucleus basalis, angular gyrus, amygdala and postcentral and supramarginal gyri) [106], was associated with depression in pwRRMS. The ACC is involved in regulating emotion, and its atrophy has been linked to anhedonia and MDD [176, 177]. Changes in choline levels within the AAC and frontal lobe have been observed in MDD and might be a potential marker for treatment outcomes in depressed patients [106, 178, 179].

*4.1.1.3 Fronto-limbic network*: *Cingulum*, *fornix and uncinate fasciculus.* Hassan et al. observed structural connectivity changes in RRMS patients with depression in the WM pathways within the fronto-limbic network, i.e., the cingulum, fornix and uncinate fasciculus [140]. The uncinate fasciculus connects the temporal lobe (containing the hippocampus and amygdala) and PFC [180]. It is involved in cognitive functioning, especially spatial and episodic verbal memory [180]. The fornix is the major pathway of the hippocampus and is associated with verbal memory [181]. The cingulum is associated with attention and executive functioning, and connects frontal, parietal, and temporal lobes. Indeed, microstructural changes in the cingulum and uncinate fasciculus were correlated with depressive symptoms in MDD [182].

*4.1.1.4 Monoamine networks.* In addition, Carotenuto et al. observed altered serotonergic-noradrenergic networks (e.g., between cerebellum and nucleus accumbens, hypothalamus, amygdala, thalamus, locus coeruleus, ventral tegmental area; brainstem and hypothalamus) in RRMS patients with depression [106]. These networks were linked to functional connectivity pathways between the cerebellum and hypothalamus, amygdala, and thalamus in depressed pwRRMS [106]. Indeed, the monoaminergic hypothesis suggests that imbalances within serotonergic-noradrenergic systems contribute to depression [183]. The serotonin network connects to cortical, limbic and brainstem regions, and is linked to the sensory, motor, or limbic systems [106, 184]. Additionally, serotonin modulates fronto-limbic circuitry in depression [185]. Meanwhile, adrenergic pathways terminate in the frontal cortex, the amygdala and the ventral striatum, and noradrenaline system controls executive functioning, cognition, and motivation [186, 187]. Loss of dopamine and noradrenaline network connectivity in the limbic system has been linked to depression in Parkinson's disease [186].

**4.1.2 Summary.** Depression in RRMS patients was mostly associated with connectivity and structural changes in cortico-limbic network, especially parts involved in fronto-limbic system: hippocampus, amygdala and PFC. This is consistent with abnormal cortical-limbic connectivity in MDD [188]. It is, however, difficult to draw firm conclusions from our study, as limited studies investigated brain connectivity underlying depression in pwRRMS. Overall, these findings suggest that clinical manifestations of depression in people with pwRRMS and MDD may have a shared biological basis, i.e., neurodegeneration in terms of myelin and axonal loss, and atrophy, of similar brain regions [94]. It would be of interest to compare brain changes in MDD with depression in pwRRMS, which may improve understanding of disease mechanisms in both conditions and could potentially lead to better treatments. Given depression is a highly common and debilitating symptom in pwRRMS [5], there is a great need for studies assessing depression in relation to MRI outcomes, particularly studies with a longitudinal design assessing brain changes underlying depression throughout the disease course.

## 4.2 Brain connectivity underlying fatigue in pwRRMS

**4.2.1 Cortico-limbic system.** *4.2.1.1 Thalamus.* Both functional [126, 130, 133] and structural connectivity [121] changes of the thalamus are associated with fatigue in pwRRMS, according to four included studies. Moreover, fatigue in pwRRMS was associated with thalamic atrophy in five studies [110, 113, 131, 148, 151], while a study by Wilting et al. found a correlation between thalamic WML volume and fatigue measures in pwRRMS [121]. This is supported by findings from Arm et al. reporting similar results for all MS subtypes [61]. Indeed, many previous studies have found the thalamus to be implicated in fatigue mixed subtype MS [189]. The thalamus controls many functions, ranging from relaying sensory and motor signals [190], as well as regulation of consciousness and alertness [191], and is also involved in cognitive functioning [192] and in regulating the sleep-wake cycle [193]. Fatigue has been previously linked to structural damage of the thalamus in post-stroke patients [194], as well as prefrontal cortex and thalamus atrophy in chronic fatigue syndrome (CFS) [195].

Structural connectivity of the anterior thalamic radiation, connecting the thalamus with the PFC and cingulate gyrus, was also found to be associated with fatigue in pwRRMS in one study [118]. This is in line with observed structural connectivity changes in thalamic radiation, which have been associated with fatigue in individuals with traumatic brain injury. These findings suggest that impaired communication between cortical and thalamic areas may contribute to the development of fatigue [196, 197].

*4.2.1.2 Frontal lobe.* The PFC showed altered functional activity and connectivity, as well as atrophy, in RRMS patients with fatigue in eleven included studies [35, 114, 119, 128–130, 133, 138, 142, 147, 152]. Part of the PFC, the DLPFC, may play a key role in fatigue in MS (not specific to RRMS). Specifically, it is part of the 'cortico-thalamo-striato-cortical loop', which has been suggested to underlie fatigue in generic MS [93, 198]. In line with these findings, previous research has found links between fatigue and DLPFC activity in healthy subjects and has also suggested the DLPFC as one of the central 'nodes' of the fatigue network in healthy individuals [199, 200]. Moreover, studies found that transcranial direct current stimulation of the DLPFC improved fatigue in (RR)MS [198, 201].

The superior (SFG), middle (MFG) and inferior (IFG) fontal gyri showed changes in functional connectivity [114, 130, 138, 152] and activation [35, 126, 128, 133] in relation to fatigue in pwRRMS, according to eight included studies. This is supported by observed SFG and MFG atrophy, as well as cortical thickness changes in the MFG [108, 110, 148]. The SFG and MFG both control working memory, but the SFG is thought to contribute to higher cognitive functions, while MFG is related to attention, especially reorienting to unexpected stimuli [202,

203]. Previously, Sepulcre et al. reported that fatigue correlated with atrophy in both the SFG and MFG in mixed subtype MS [204]. Additionally, IFG is implicated in processes associated with attention and task-switching functions [205] and has been linked to CFS [206].

Functional connectivity changes were also observed in brain motor areas in ten included studies [35, 114, 126–128, 133, 138, 148, 152, 153]. The premotor cortex plays a role in motor fatigue specifically, in healthy individuals [207], and is involved in planning and organizing movements and actions [208]. Furthermore, SMA contributes to the simple motor control and pre-SMA is involved in complex cognitive and motor control [209, 210]. Both SMA and pre-SMA showed changes in activation due to fatigue, with the former being more activated in motor fatigue especially [210]. Additionally, fatigue in pwRRMS was also found to be associated with functional changes in the pre- and postcentral gyrus of the SMN, controlling voluntary motor movement and proprioception, respectively [211]. This is supported by previously observed decreased functional activity of the precentral cortex [212, 213] in CFS [195]. Similarly, functional connectivity of the postcentral gyrus was also affected in CFS [213].

*4.2.1.3 Parietal and temporal lobes.* Functional connectivity changes of the supramarginal gyrus and precuneus were both associated with fatigue in pwRRMS in six included studies [114, 126, 127, 133, 138, 152]. In line with this, reduced functional connectivity of the supramarginal gyrus and postcentral gyrus was associated with fatigue in CFS [213], and FC in supramarginal gyrus was associated with fatigue in traumatic brain injury [214]. The supramarginal gyrus is a part of the frontoparietal network, and plays a role in attention, verbal working memory and emotional responses [214–216]. The precuneus, on the other hand, is involved in higher-order neurocognitive processes, including motor coordination, mental rotation, and episodic memory retrieval [217]. Indeed, Chen et al. previously reported that cognitive fatigue in generic MS was correlated with reduced functional connectivity of the precuneus [218].

Fatigue scores were associated with altered functional connectivity of temporal gyri in four studies [35, 115, 133, 138], and reduced FA in the R-temporal lobe [123]. Moreover, atrophy and lesion studies have shown temporal lobe atrophy and white matter lesions in fatigue pwMS [108, 219]. The role of the temporal lobes in fatigue is further supported by previous literature showing temporal lobe involvement in fatigue in Parkinson's disease [220, 221].

*4.2.1.4 Cingulum and cingulate gyrus.* Fatigue in pwRRMS was associated with structural [117] and functional connectivity changes [114, 126, 133, 152], as well as atrophy, in the cingulate cortex [108, 148] in seven included studies. It is a key component of the limbic system [222], and is involved in processing emotions and behaviour regulation [223]. Indeed, previous research associated abnormal functional connectivity change of the cingulate with fatigue in CFS [224]. The cingulate gyrus is closely connected to the cingulum, which links it with subcortical nuclei [225]. Both structural and functional connectivity changes of the cingulum were associated with fatigue [117, 118, 133]. The cingulum is a prominent WM tract required for motivational processes, mood modulation, and emotion recognition [118]. Previously, a link between lesions in the cingulum and fatigue has been observed in mixed subtype MS [204]. Additionally, fatigue in Parkinson's disease was correlated with altered functional connectivity in the cingulum [226].

**4.2.2 Basal ganglia.** Basal ganglia regions also play a role in fatigue symptomatology in pwRRMS, as both structural [121] and functional connectivity [35, 110, 114, 123, 128–130, 133, 138, 145, 155] changes, and atrophy [35, 108, 110, 113, 114, 128], were observed in RRMS patients with fatigue. The basal ganglia nuclei are primarily responsible for motor control, motor learning, executive functions, and behaviours, as well as emotions [227]. Previous research by Nakagawa et al. suggested that abnormal function of the motor and dopaminergic system in the basal ganglia, which are associated with motivation and reward, are underlying

fatigue in CFS [228]. This is further supported by basal ganglia changes in association with fatigue in Parkinson's disease and in healthy subjects [229].

Abnormal activation of basal ganglia has also been observed in fatigued RRMS patients [128, 133] by two included studies. This is in line with healthy ageing research showing that cortico-striatal networks play a role in fatigue [230]. The striatum (a basal ganglia nuclei) is associated with cognitive control and motivation [231], both functions related to fatigue [232]. A key WM tract in the fronto-striatal network is the inferior fronto-occipital fasciculus, which has shown structural connectivity abnormalities in fatigue in pwRRMS [118]. In support of this, inferior fronto-occipital fasciculus atrophy has been observed in people with CFS [233]. Interestingly, previous research has suggested the dopamine imbalance hypothesis, which supposes that fatigue arises due to a dopamine imbalance within the fronto-striatal network in pwRRMS [234]. Furthermore, it has also been suggested that connectivity changes in this tract may negatively affect the integration of sensory information and inhibition control over impulses and emotion [235], leading to fatigue.

Another WM tract important for basal ganglia functioning is the internal capsule. It connects the basal ganglia with the limbic network [107, 236] and contributes to physical movement and perception of sensory information [237]. Here, we observed that structural connectivity of the internal capsule was associated with fatigue scores in pwRRMS [118]. This is supported by previous findings showing reduced white matter microstructural integrity of the internal capsule in fatigue in traumatic brain injury [237].

**4.2.3 Cerebellum.** Functional connectivity of the cerebellum was associated with fatigue scores in pwRRMS in two included studies [128, 133]. This is supported by two other studies associating fatigue and cerebellar atrophy [139, 148]. In line with this, cerebellar lesion volume was identified as an independent predictor of fatigue in pwRRMS [113]. Similarly, cerebellar volume has previously been found to predict fatigue severity changes in early MS [238]. The cerebellum plays a critical role in sensorimotor behaviour, automation [239] and cognitive tasks [139]. Indeed, increased activation in cerebellum in mixed subtype MS was linked to cognitive fatigue during a task-switching task [240] and changes in cerebellar activity in healthy volunteers were associated with a motor fatigue in fMRI study [210].

**4.2.4 Default mode network.** Reduced activity [115] and dynamic functional connectivity [155] in the DMN was associated with fatigue. This was supported by structural connectivity changes and atrophy in regions of the DMN observed in fatigued pwRRMS [147, 148]. The DMN is involved in emotional processing, memory and task performance [241, 242], and the observed link between altered DMN connectivity and fatigue in pwMS may potentially be due to microstructural damage, or rearrangement of networks to compensate for DMN dysfunction [91, 243]. The role of the DMN changes underlying fatigue is further supported by reported associations of fatigue and increased activation of the DMN in CFS patients [244] and DMN hyperconnectivity in breast cancer survivors [243].

**4.2.5 Summary.** The existing literature indicate that structural and functional changes in regions of the cortico-thalamocortical and cortical-subcortical circuits are associated with fatigue. There seems to be an overlap of different MRI measures relating to fatigue in thalamus, basal ganglia, cingulum, cerebellum, cingulate cortices, motor areas, and regions in the frontal, temporal, and parietal lobes in patients with RRMS. Most of these regions are thought to be involved in motor and cognitive functions as well as reward seeking behaviour which fatigue has been previously shown to affect [245–247]. Overall, these results suggest a link between fatigue and neurodegenerative processes in specific areas of the brain. The similarities between brain changes associated with fatigue in pwRRMS and other disorders suggest that damage to distinct structures could lead to development of fatigue. It may also indicate a possibility for shared treatments such as cognitive behavioural therapy, cryotherapy, and balance and/or

multicomponent exercise, both of which show promising results in CFS [248–250]. However, about half of the literature in this review reported negative findings, and the positive findings were highly variable.

## 4.3 Overlapping brain connectivity changes associated with depression and fatigue in pwRRMS

Depression and fatigue are interlinked and overlap in symptomatology [9, 51], making it difficult to differentiate between them in pwRRMS. This is further complicated by the multidimensional nature of fatigue and the influence of factors such as sleep disturbance and neuropathic pain on both depression and fatigue in people with mixed subtype MS [251]. Previous literature investigating associations between depression and fatigue in people with any subtype of MS have given disparate results, but with consensus that there is some association between them [252, 253]. The current review indeed suggests that there may be overlap in brain changes underlying fatigue and depression in pwRRMS. Specifically, structural connectivity in cingulum and functional connectivity in cerebellum, thalamus, PFC, supramarginal gyrus, ventral tegmental area, superior ventral striatum, DMN, attention networks, and pre/post-central gyri. There is ample evidence of these regions' involvement in both depression and fatigue [61, 117, 118, 139, 191, 218, 225, 234, 243, 254–262]. However, FC changes included in this study displayed heterogeneity, likely due to differences in study design, methodology, and disease stage. Both depression and fatigue were associated with connectivity changes in the cortico-limbic network, and especially the fronto-limbic network. However, especially due to limited studies investigating depression in pwRRMS, more research is needed to pinpoint the underlying mechanisms driving these comorbidities.

## 4.4 Limitations of studies included in the systematic review

First, studies included in the review were heterogeneous in methodology, particularly around study design, fatigue, and depression assessments (e.g., inclusion/exclusion criteria), imaging protocols (including different MRI systems and strengths), sample size, and reporting of results. Furthermore, studies used different data processing protocols and statistical analysis approaches. Lack of standardisation in acquisition and imaging processing methods significantly reduces the ability of researchers to combine data meaningfully from different studies. Such differences make it difficult to formally compare studies and replication studies are needed.

Secondly, the innate and complex interaction and overlap between fatigue and depression limits interpretation of the findings. We tried to limit the variation by only including depression and fatigue assessments validated in MS and by focusing on the most 'popular' imaging techniques. As depression and fatigue are multifaceted disorders, with variable symptoms and manifestations, separating the symptoms by their function (as was done for motor/cognitive fatigue) could help to clarify the issue in the future. Moreover, many studies assessing the link between MRI outcomes and fatigue did not consider depression status—and vice versa. This makes it challenging to attribute findings to one symptom alone, especially as depression and fatigue are so intertwined.

Moreover, some studies were focusing on regions previously associated with depression and fatigue in RRMS, thus, potentially overlooking other significant parts of the brain.

Similarly, very few studies investigated both symptoms together, preventing any firm conclusions to be drawn on shared disease mechanisms in the brain between fatigue and depression in pwRRMS. Therefore, overlapping results are based on comparing study outcomes for fatigue and depression separately. This illustrates the lack of research on the link between

depression and fatigue in pwRRMS and indicates future research should focus on further elucidation of underlying disease mechanisms for both symptoms combined, particularly using advanced imaging methods that allow for detection of more subtle brain changes.

## 4.5 Limitations of this study

The scope of our review was limited, resulting from database screenings done without citation mapping. We expect, however, that as three databases were explored, most relevant literature has been covered. Only publications in English were considered, which may mean some findings have been missed. Additionally, studies assessing the effects of drug treatments were excluded and hence relevant information potentially unrelated to the therapy may have been missed. Furthermore, in order to reduce possibly incorrect conclusions based on samples with low numbers of participants, we chose a cut-off value of $\geq$20. Although we realise this is an arbitrary threshold, we had to balance between excluding too few or too many papers. We also only focused on brain connectivity using dMRI and fMRI and did not consider other microstructural or physiological imaging methods (e.g., magnetisation transfer imaging, MR spectroscopy, or positron emission tomography). Moreover, there were very few studies using NODDI or PSMD, and none met our inclusion criteria. Future reviews should include such measures to further elucidate common mechanisms for fatigue and depression in pwRRMS.

Additionally, we did not include spinal cord imaging studies given the relative lack of studies investigating spinal cord connectivity, likely due to technical limitations [263]. Furthermore, we did not formally assess publication bias, however, aimed to provide a complete overview of positive and negative outcomes. Lastly, qualitative approach prevents accurately assessing the strength of interactions. The studies included in this review used standard statistical significance cut-off values, and where correlations were statistically significant, they tended to be weak. In the future, a rigorous quantitative analysis could elucidate the heterogeneity of the current results.

## 4.6 Conclusion

Overall, the results presented were highly variable; half of those reviewed found no significant associations between brain connectivity measures and depression or fatigue. Studies reporting positive findings showed that a) brain connectivity and macrostructural changes in the cortico-thalamic-basal ganglial network were associated with fatigue in pwRRMS, b) cortico-limbic networks were associated with depression in pwRRMS, and c) structural connectivity in the cingulum and functional connectivity in the cerebellum, thalamus, frontal lobe, supramarginal gyrus, ventral tegmental area, superior ventral striatum, DMN, attention networks, and pre/post-central gyri was affected in both fatigue and depression in pwRRMS. This may suggest that structural damage of WM and GM (e.g., neuroaxonal loss and/or demyelination) within these regions is responsible for depression and fatigue in pwRRMS, albeit not consistent findings across the literature. These mixed findings are most likely due to heterogeneous methodology across the studies. Only a small number of studies investigated brain connectivity in depression, or in both depression and fatigue combined. Moreover, the complex relationship and overlap between these two phenomena complicates interpretation of findings. Further adequately powered studies using optimised structural, microstructural, and functional imaging measures in well-characterised RRMS cohorts with validated indices of fatigue and depression are required to determine jointly affected brain areas in depression and fatigue, and further elucidate disease mechanisms underlying these symptoms. Moreover, studies employing additional imaging modalities such as positron emission tomography (PET) could be reviewed to further investigate the relationship between brain changes and fatigue/depression in pwRRMS.

## Supporting information

**S1 Checklist. The PRISMA 2020 main checklist filled in for the current systematic review.**
(PDF)

**S2 Checklist. The PRIMSA abstract checklist filled in for the current systematic review.**
(PDF)

**S1 Text. Supplement methods and supplement results.**
(PDF)

**S1 Table. An overview of all studies read in full and final decision.**
(PDF)

**S2 Table. Data extraction table.**
(PDF)

**S3 Table. Quality assessment of cross-sectional studies using the 'Appraisal tool of cross-sectional studies' (AXIS).**
(PDF)

**S4 Table. Quality assessment of longitudinal studies using the Institute of Health Economics (IHE) 'Quality appraisal of case series studies checklist'.**
(PDF)

**S5 Table. Full breakdown of AXIS scores.**
(PDF)

**S6 Table. Full breakdown of Institute of Health Economics (IHE) checklist scores.**
(PDF)

**S7 Table. Overview of study details for publications included (N = 60) in the current systematic review.**
(PDF)

**S8 Table. Negative findings explicitly reported in the included studies of this systematic review.**
(PDF)

**S9 Table. Literature search results.**
(PDF)

## Acknowledgments

For open access, the author has applied a Creative Commons Attribution (CC BY) licence to any Author Accepted Manuscript version arising from this submission.

## Author Contributions

**Conceptualization:** Elizabeth N. York, Siddharthan Chandran.

**Writing – original draft:** Agniete Kampaite, Rebecka Gustafsson, Adam D. Waldman, Rozanna Meijboom.

**Writing – review & editing:** Elizabeth N. York, Peter Foley, Niall J. J. MacDougall, Mark E. Bastin, Siddharthan Chandran, Adam D. Waldman, Rozanna Meijboom.

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
