## [Decision Letter · Decision Letter 0]

14 Jul 2023

PONE-D-23-09587Brain connectivity changes underlying depression and fatigue in relapsing-remitting multiple sclerosis: a systematic reviewPLOS ONE

Dear Dr. Kampaite,

Thank you for submitting your manuscript to PLOS ONE. After careful consideration, we feel that it has merit but does not fully meet PLOS ONE’s publication criteria as it currently stands. Therefore, we invite you to submit a revised version of the manuscript that addresses the points raised during the review process.

We look forward to receiving your revised manuscript.

Kind regards,

Francesca Benuzzi, Ph.D.

Academic Editor

PLOS ONE

Journal Requirements:

Additional Editor Comments :

Dear Authors,

both Reviewers found the review very interesting and particularly relevant to the filed. However, they found major issues to be addressed before the paper should be ready for publication. In particular, the most critical point is the rationale to include depression and fatigue and the relationship that occurs between the two.

Another relevant point is the description of the methodology used to extract and code of data that need more clarification.

I am confident that the points raised by Revierws may be addressed and resolved thus making the paper suitable for publication.

Regards,

Reviewers' comments:

Reviewer's Responses to Questions

**Comments to the Author**

1. Is the manuscript technically sound, and do the data support the conclusions?

Reviewer #1: Yes

Reviewer #2: Yes

2. Has the statistical analysis been performed appropriately and rigorously? 

Reviewer #1: N/A

Reviewer #2: N/A

3. Have the authors made all data underlying the findings in their manuscript fully available?

Reviewer #1: Yes

Reviewer #2: Yes

4. Is the manuscript presented in an intelligible fashion and written in standard English?

Reviewer #1: Yes

Reviewer #2: Yes

5. Review Comments to the Author

Reviewer #1: Please check the attached.

The PRISMA approach clearly shows the search process. The brain network reviews for depression and fatigue were a strength of the paper. The limitations of the studies included in the paper serve as a valuable resource for researchers to identify research gaps, which is one of the key objectives of a literature review. It was a nice job!

However, there are a few points that could be modified and added:

1) It is unclear why depression and fatigue were reviewed together and how they are related. The authors could have provided a rationale for why other MS cognitive symptoms were not included.

2) The sentence "This is supported the observation that depression is more prevalent in MS than in other neurodegenerative disorders" could be improved by providing a reference that compares MS to other neurodegenerative disorders in this regard. With this references, It is not proven that depression is more prevalent in MS than in ALS, for instance.

3) The authors utilized three top databases in their search for relevant literature; however, it should be noted that Google Scholar results are also a comprehensive resource when conducting a systematic review. (Please check this paper if it is eligible to be included , found from first page of search on Google scholar with the inclusion and exclusion criteria.)

• Carotenuto, A., Valsasina, P., Preziosa, P., Mistri, D., Filippi, M., & Rocca, M. A. (2023). Monoaminergic network abnormalities: a marker for multiple sclerosis-related fatigue and depression. Journal of Neurology, Neurosurgery & Psychiatry, 94(2), 94-101.

4) The research questions are not stated in the methodology section. The authors should perfectly illustrate the aim of the study.

5) The use of "PwRRMS" as a short form is not common and could be confusing in the paper.

6) Table 1 does not need to be in the main text and could be summarized in a few lines.

7) In Table 4, the "Brain Region" column does not necessarily represent the region. For example, in cells 1-3 I, it shows the condition rather than regions.

8) The authors presented the overlapping of depression and fatigue in the papers included in the review using three long tables. It would be helpful for the readers to better understand the relationship between depression and fatigue in MS patients and to identify potential research gaps. In the text, it would be beneficial to provide a brief explanation of why it is important to study the co-occurrence of these symptoms.

9) The authors discussed brain networks perfectly and in a well-structured manner. The same could be done for disease progression regarding depression and fatigue in MS patients.

Reviewer #2: Kampaite and colleagues propose an interesting review aiming to elucidate the relationship between depression and fatigue and brain connectivity in patients with relapsing-remitting Multiple Sclerosis (MS). Their searches were conducted using three databases (PubMed, Web-of-Science and Scopus). Studies employing fatigue and depression assessments validated for MS and included brain structural, functional magnetic resonance imaging (fMRI) or diffusion MRI (dMRI) were included. After selection process, Sixty studies met the criteria: 18 diffusion MRI (dMRI) (15 fatigue, 5 depression) and 22 functional MRI (fMRI) (20 fatigue, 5 depression) studies were included.

They found a heterogeneous literature since half of studies reported no correlation between brain connectivity measures and fatigue or depression. Positive findings showed that abnormal cortico-limbic structural and functional connectivity was associated with depression. Fatigue was linked to connectivity measures in cortico-thalamic-basal- ganglial networks. Additionally, both depression and fatigue were related to altered cingulum structural connectivity, and functional connectivity involving thalamus, cerebellum, frontal lobe, precentral and postcentral gyri.

Qualitative analysis suggests neuropathological effects, possibly due to axonal and/or myelin damage, in the cortico-thalamic-basal-ganglial and cortico-limbic network may underlie fatigue and depression in patients with MS, respectively, but the overall results were inconclusive.

The review is relevant for MS research community and well conducted.

However, some revisions may improve the quality and readability of the manuscript.

Studies assessing the effects of disease modifying therapies were excluded. According to the PRISMA flowchart, seven studies were excluded due to this criterion. However, I wonder if a baseline evaluation could have been employed.

Authors had to register their review proposal in PROSPERO that represents an online database created to provide a comprehensive listing of systematic reviews registered at inception to help scientific community to avoid duplication.

Negative findings were reported by half of the studies included in the systematic review; did the authors try to qualitatively analyze in which way studies reporting negative findings were different from those that reported significant associations?

Authors should better describe the methodology behind extraction and coding of data for descriptive purposes (Table and population characteristics) and outcome discussion. For example, how many authors participated in this phase? Did they work in blind mode? How did they reach an agreement about data to be entered in case of disagreement?

I would suggest, where possible, to make sections/paragraphs more coincise in order to make the article more readable (e.g. it would be better to underpin converging/overlapping evidences more than describing/reporting the results of any single significant study - that are comprehensively reported/mentioned in the Tables)

Other/Minor revisions:

I would always mention the number of significant studies out of the total availbale studies with a specific MRI metric/methodology (e.g. 3 RS-FMRI studies found significant results out of 6 available RS-FMRI studies)

rows 374-380: which is/are the considered MRI measure/s (atrophy, volume, CTh, WML load)?

row 466-470: check the numbers (it seems that the sum of the reported numbers do not does 29)

In Table 3 I would better specify the significant MRI measure (i.e. atrophy, CTH, diffusion measures, etc)

Why TBSS measures were not considered?

Why CTh and VBM atrophy patterns were not considered?

6. PLOS authors have the option to publish the peer review history of their article (what does this mean?). If published, this will include your full peer review and any attached files.

Reviewer #1: **Yes: **Sara Hejazi

Reviewer #2: No

---

## [Author Response · Author response to Decision Letter 0]

7 Oct 2023

We thank the reviewers for their helpful comments and queries, which we have addressed on a point-by-point basis below, and referenced changes to the manuscript in response to these.

Journal Requirements:

We apologise for having overlooked any requirements. We have now carefully checked the manuscript for matching PLOS One’s style reference and have made modifications where required.

We apologise for the error; the correct funding is as outlined in the manuscript. We have also included the funding section below for the editor’s convenience.

“Funding for authors came from the MS Society Edinburgh Centre for MS Research (grant reference 133; AK, RM), Chief Scientist Office – SPRINT MND/MS program (grant reference MMPP/01; ENY), the Anne Rowling Regenerative Neurology Clinic (ENY), and the UK Dementia Research Institute which receives its funding from UK DRI Ltd, funded by the UK Medical Research Council, Alzheimer’s Society and Alzheimer’s Research UK (SC).”

We apologise for any confusion. Our systematic review does not contain any raw data, however, we have now ensured that all documents created in the article selection are included in the supplement and hence available to the public. We have now disclosed all the relevant data and made a reference to these documents in the methods section of the manuscript. 

“See S1 Table for an overview of all studies reviewed and their selection process.”

Additional Editor Comments:

E1. The most critical point is the rationale to include depression and fatigue and the relationship that occurs between the two.

We agree with the editor that the rationale should be more clearly elaborated. We chose to focus our systematic review on fatigue and depression, because they are highly prevalent in people with MS, may be causally interrelated, have overlapping features and often co-occur. Moreover, there is a lack of understanding why these two conditions affect MS patients particularly and current treatment options are limited. We have now modified the introduction accordingly and hope it is sufficiently clear.

“Higher prevalence of depression in MS than in the general population has been previously reported [8], and fatigue may affect 60-80% of people with newly diagnosed MS [9]. Both fatigue and depression are associated with decreased quality of life in people with MS [10] and are considered major debilitating symptoms [11], together affecting more than 50% of people with MS [10]. The relationship between depression and fatigue is complex; although considered distinct entities, there is a high degree of comorbidity and their phenotypes overlap (e.g., anhedonia, sleep disturbance) [12, 13]. Fatigue is considered both a symptom and a consequence of depression, and conversely, people with fatigue are more likely to report depressive symptoms [13, 14]. Associations of fatigue and depression and other MS symptoms, such as pain, cognition, and anxiety have also been found [15-21]. In view of the strong overlap of fatigue and depression, however, this review will focus on establishing a better understanding of the substrate for fatigue and depression, and their relationship to known MS pathobiology.”

E2. Another relevant point is the description of the methodology used to extract and code of data that need more clarification. 

We appreciate that this was not clear. Data were extracted by one reviewer, using a custom-made template tailored for this review. We have now added further clarification on the methodology to Methods section 2.2. 

“Final selections were compared to reach consensus. In case of a disagreement, the reviewers re-read the paper and either amended their decision or made further arguments for their initial choice. Persisting discrepancies were discussed together with a third reviewer and final decisions were made by consensus.

The data was extracted by one reviewer into a standardised table designed for this review.” 

Reviewers’ comments:

Reviewer #1: Please check the attached.

R1.1) It is unclear why depression and fatigue were reviewed together and how they are related. The authors could have provided a rationale for why other MS cognitive symptoms were not included.

We agree with the reviewer that the rationale should be more clearly elaborated. This point was also raised by the Editor and is addressed in our response to them. We chose to focus our systematic review on fatigue and depression, because they are highly prevalent in people with MS, may be causally interrelated, have overlapping features and often co-occur. Moreover, there is a lack of understanding why these two conditions affect MS patients particularly and current treatment options are limited. We have now modified the introduction accordingly and hope it is sufficiently clear.

Fatigue and depression show a strong overlap and are highly prevalent in MS. We were interested in identifying underlying brain changes contributing to this overlap. Other features of MS, such as cognitive impairment, do not show the same degree of overlap and may have different underlying mechanisms, so are beyond the scope of this review. 

“Higher prevalence of depression in MS than in the general population has been previously reported [8], and fatigue may affect 60-80% of people with newly diagnosed MS [9]. Both fatigue and depression are associated with decreased quality of life in people with MS [10] and are considered major debilitating symptoms [11], together affecting more than 50% of people with MS [10]. The relationship between depression and fatigue is complex; although considered distinct entities, there is a high degree of comorbidity and their phenotypes overlap (e.g., anhedonia, sleep disturbance) [12, 13]. Fatigue is considered both a symptom and a consequence of depression, and conversely, people with fatigue are more likely to report depressive symptoms [13, 14]. Associations of fatigue and depression and other MS symptoms, such as pain, cognition, and anxiety have also been found [15-21]. In view of the strong overlap of fatigue and depression, however, this review will focus on establishing a better understanding of the substrate for fatigue and depression, and their relationship to known MS pathobiology.”

R1.2) The sentence "This is supported the observation that depression is more prevalent in MS than in other neurodegenerative disorders" could be improved by providing a reference that compares MS to other neurodegenerative disorders in this regard. With this references, It is not proven that depression is more prevalent in MS than in ALS, for instance.

We agree with the reviewer and have now added five references (see below) to add support to the notion that depression may be more prevalent in MS than in other neurodegenerative disorders. Additionally, we have modified our statement.

“This is supported by the observation that depression may be more prevalent in MS than in other neurodegenerative/inflammatory disorders [30-34].”

New references:

1. Lobentanz, I. S., et al. (2004). "Factors influencing quality of life in multiple sclerosis patients: disability, depressive mood, fatigue and sleep quality." Acta Neurol Scand 110(1): 6-13.

2. Taylor, L., et al. (2010). "Prevalence of depression in amyotrophic lateral sclerosis and other motor disorders." Eur J Neurol 17(8): 1047-1053.

3. Tauil, C. B., et al. (2018). "Suicidal ideation, anxiety, and depression in patients with multiple sclerosis." Arq Neuropsiquiatr 76(5): 296-301.

4. Schiffer, R. B. and H. M. Babigian (1984). "Behavioral disorders in multiple sclerosis, temporal lobe epilepsy, and amyotrophic lateral sclerosis. An epidemiologic study." Arch Neurol 41(10): 1067-1069.

5. Minden, S. L., et al. (1987). "Depression in multiple sclerosis." General Hospital Psychiatry 9(6): 426-434.

R1.3) The authors utilized three top databases in their search for relevant literature; however, it should be noted that Google Scholar results are also a comprehensive resource when conducting a systematic review. (Please check this paper if it is eligible to be included, found from first page of search on Google scholar with the inclusion and exclusion criteria.)

• Carotenuto, A., Valsasina, P., Preziosa, P., Mistri, D., Filippi, M., & Rocca, M. A. (2023). Monoaminergic network abnormalities: a marker for multiple sclerosis-related fatigue and depression. Journal of Neurology, Neurosurgery & Psychiatry, 94(2), 94-101.

While we agree with the reviewer that Google Scholar is a good database, we do not, believe it is any better or worse than those we chose to use here. We included three established and widely-used databases in our search and are confident this allows for sufficient coverage of the literature. 

We thank the reviewer for suggesting the above paper; the publication was identified in the initial search output, but was deemed not eligible, as it does not report results for RRMS and progressive MS separately. 

R1.4) The research questions are not stated in the methodology section. The authors should perfectly illustrate the aim of the study.

We believe that the aims were elaborated clearly in Introduction section 1.4. The relevant section is reproduced below for the reviewer’s convenience. We have edited wording of the aims paragraph to be more succinct. If the reviewer feels that the rationale aims are still unclear, please could we respectfully ask them to be more specific about what should be included? 

“The aim of this study is to systematically review the literature to elucidate the relationship between structural and brain connectivity MRI measures and depression or fatigue in pwRRMS. This may provide new insights into brain changes in RRMS related to depression and fatigue.” 

R1.5) The use of "PwRRMS" as a short form is not common and could be confusing in the paper.

We respectfully disagree with the reviewer. ‘pwRRMS’ has been used in other papers [6-8]. Indeed, the abbreviation has been published in PLOS ONE before [8]. We also believe this is a more respectful way to refer to patients/study participants with MS. For these reasons, we would prefer to maintain the use of this abbreviation in the manuscript.

References:

6. Dacosta-Aguayo R, Genova H, Chiaravalloti ND, DeLuca J. Neuroimaging and Rehabilitation in Multiple Sclerosis. In: DeLuca J, Chiaravalloti ND, Weber E, editors. Cognitive Rehabilitation and Neuroimaging: Examining the Evidence from Brain to Behavior. Cham: Springer International Publishing; 2020. p. 117-38.

7. Habek M. Immune and autonomic nervous system interactions in multiple sclerosis: clinical implications. Clinical Autonomic Research. 2019;29(3):267-75. doi: 10.1007/s10286-019-00605-z.

8. Lamargue Hamel D, Deloire M, Ruet A, Charré-Morin J, Saubusse A, Ouallet JC, et al. Deciphering Depressive Mood in Relapsing-Remitting and Progressive Multiple Sclerosis and Its Consequence on Quality of Life. PLoS One. 2015;10(11):e0142152. Epub 20151110. doi: 10.1371/journal.pone.0142152. PubMed PMID: 26555230; PubMed Central PMCID: PMCPMC4640551.

R1.6) Table 1 does not need to be in the main text and could be summarized in a few lines.

We have removed the table and summarised its contents in the text instead.

“Studies were included if they met the following inclusion criteria: (1) structural, diffusion or functional MRI was used to study brain changes, (2) included a minimum sample size of 20 participants, (3) assessed either RRMS alone or distinguished between MS subtypes, and (4) used depression or fatigue assessments validated for use in MS, based on three previous reviews of MS-related depression and fatigue [1, 4, 5](5-7) (Depression assessment tools: Beck Depression Index (BDI)([82), Beck Depression Index-II (BDI-II) (83), Diagnostic and Statistical Manual V semi-structured interview (DSM-V) (84), Centre for Epidemiological Studies – Depression (CES-D) (84), Chicago Multiscale Depression Inventory (CMDI) (84), Patient Health Questionare-9 (PHQ-9) (84), Hospital Anxiety and Depression Scale (HADS) (87), Hamilton Depression Rating Scale (HDRS) (88); Fatigue assessment tools: Fatigue Severity Scale (FSS) (29), Modified Fatigue Impact Scale (MFIS) (29), Fatigue Impact Scale (FIS) (85), Fatigue Scale for Motor and Cognitive functions (FSMC) (31), Checklist of Individual Strength (CIS-20r) (86). Short descriptions for each measure can be found in Gümüş (85) or Cheung (89)). Studies were excluded…”

R1.7) In Table 4, the "Brain Region" column does not necessarily represent the region. For example, in cells 1-3 I, it shows the condition rather than regions.

We agree with the reviewer and now have changed the column name to ‘Brain region/network’ to be more accurate. We have also updated Table 5 accordingly.

R1.8) The authors presented the overlapping of depression and fatigue in the papers included in the review using three long tables. It would be helpful for the readers to better understand the relationship between depression and fatigue in MS patients and to identify potential research gaps. In the text, it would be beneficial to provide a brief explanation of why it is important to study the co-occurrence of these symptoms.

We agree with the reviewer and have added further explanation to the introduction. We would also like to refer the reviewer to our answer to comment R1.1. 

R1.9) The authors discussed brain networks perfectly and in a well-structured manner. The same could be done for disease progression regarding depression and fatigue in MS patients.

We apologise for not fully comprehending the reviewer’s comment.

The relationship between depression and MS disease progression, and equally between fatigue and MS disease progression, was not the aim of this study. We agree with the reviewer, however, that this would be of great interest for further study. In this study, we focussed on relationships of brain structural and network measures with fatigue/depression in MS. We have edited wording of the aims paragraph to be more succinct, and make this clearer. We would also like to refer the reviewer to our answer to comment R1.4. 

We are not entirely certain we have answered the reviewer’s question correctly, for which we apologise. We would respectfully like to ask whether the reviewer is able to provide additional clarification if we have not sufficiently responded to their comment. 

 

Reviewer #2: 

R2.1. Studies assessing the effects of disease modifying therapies were excluded. According to the PRISMA flowchart, seven studies were excluded due to this criterion. However, I wonder if a baseline evaluation could have been employed.

We aimed to avoid any confounding effects of treatments on brain changes underlying fatigue and depression, and therefore excluded studies specifically evaluating the effect of treatments. The reviewer rightly highlights the potential inclusion of baseline evaluations; however, it is worth noting that a significant portion of these studies either did not involve drug-naïve patients, utilized a limited control group (n<20), or did not meet other criteria in our review. Our decision to narrow our focus was driven by the expectation that the added value of this data would have been minimal. Furthermore, drug studies specifically assessing associations of fatigue and depression with underlying brain connectivity changes while on a particular treatment (and as such possibly providing a control group without treatment) are highly limited. Out of 65 treatment studies included after our initial search, only one study [8] met our inclusion criteria (while ignoring the treatment exclusion criteria). They observed that modified fatigue impact scale (MFIS) scores correlated with resting state functional connectivity (rs-FC) in the anterior cingulate cortex, and inversely correlated with rs-FC in the thalamus [8]. We have discussed the relationship between these two brain regions and fatigue in our review, and, therefore, including this study would not alter our conclusions. Considering only 1 paper could have been added, we would suggest to maintain our current drug trial exclusion criteria. 

Reference:

9. Rocca MA, Valsasina P, Colombo B, Martinelli V, Filippi M. Cortico-subcortical functional connectivity modifications in fatigued multiple sclerosis patients treated with fampridine and amantadine. Eur J Neurol. 2021;28(7):2249-58. Epub 20210426. doi: 10.1111/ene.14867. PubMed PMID: 33852752.

R2.2. Authors had to register their review proposal in PROSPERO that represents an online database created to provide a comprehensive listing of systematic reviews registered at inception to help scientific community to avoid duplication.

We agree with the reviewer. Indeed this brilliant initiative promotes greater transparency in research practices. Regrettably, we were unaware of PROSPERO when commencing this review. We will ensure proposal registration through PROSPERO for any future reviews.

R2.3. Negative findings were reported by half of the studies included in the systematic review; did the authors try to qualitatively analyze in which way studies reporting negative findings were different from those that reported significant associations?

We have indeed examined differences between papers, however, the majority of studies reported both positive and negative findings. We therefore conclude no differences were present between studies reporting positive or negative findings.

R2.4. Authors should better describe the methodology behind extraction and coding of data for descriptive purposes (Table and population characteristics) and outcome discussion. For example, how many authors participated in this phase? Did they work in blind mode? How did they reach an agreement about data to be entered in case of disagreement?

We agree with the reviewer and have now added any missing details to section 2.2. This section is also included below for the reviewers’ convenience. In brief, two reviewers independently searched the databases and selected relevant papers; followed by a discussion to reach consensus. Any persisting discrepancies were discussed with a third reviewer, to make a final decision. Finally, one reviewer extracted study characteristics from the final selection of papers. This reviewer was not blinded to the studies. 

“Final selections were compared to reach consensus. In case of a disagreement, the reviewers re-read the paper and either amended their decision or made further arguments for their initial choice. Persisting discrepancies were discussed together with a third reviewer and final decisions were made by consensus. The data was extracted by one reviewer into a standardised table designed for this review.”

R2.5. I would suggest, where possible, to make sections/paragraphs more coincise in order to make the article more readable (e.g. it would be better to underpin converging/overlapping evidences more than describing/reporting the results of any single significant study - that are comprehensively reported/mentioned in the Tables)

We agree with the reviewer and have appropriately condensed the sections and paragraphs as necessary.

Other/Minor revisions:

1. I would always mention the number of significant studies out of the total availbale studies with a specific MRI metric/methodology (e.g. 3 RS-FMRI studies found significant results out of 6 available RS-FMRI studies)

We agree with the reviewer and have now added the number of total available studies where appropriate.

2. rows 374-380: which is/are the considered MRI measure/s (atrophy, volume, CTh, WML load)?

We would like to thank the reviewer for identifying this error. We have now added the appropriate MRI measures to the text. 

3. row 466-470: check the numbers (it seems that the sum of the reported numbers do not does 29)

Indeed, the discrepancy in numbers arises from papers that incorporated depression assessments, which excluded individuals with high depression scores and simultaneously controlled for depression status. We have now revised the sentence in the text to make this clearer.

“Of the 41 publications reporting results of MRI measures in relation to fatigue only, 29/41 included depression assessments [34, 103-106, 109, 113-118, 121, 122, 125-130, 145-150], with 14/29 excluding participants with high depression scores [34, 103, 104, 109, 113, 116-118, 121, 123, 125, 126, 128, 130] and 11/29 controlling for depression status [105, 106, 108, 122, 127, 129, 143-145, 149, 150], or both (5/29) [114, 115, 146-148].”

4. In Table 3 I would better specify the significant MRI measure (i.e. atrophy, CTH, diffusion measures, etc)

We have now specified the appropriate MRI measures in Tables 3-5. Additionally, we have made significant revisions to the functional connectivity table (Table 4 in the updated manuscript) in response to the reviewer's feedback and have ensured that it is now presented in a more informative manner. We have expanded and restructured the table to ensure that it encompasses all relevant networks and connections linked to depression and fatigue, and clearly states all overlapping areas. Additionally, we have updated the corresponding text in the manuscript to provide a clearer explanation of the functional connectivity results.

5. Why TBSS measures were not considered?

We apologise for any previous lack of clarity in our methodology. We included the general search terms 'diffusion MRI' and 'diffusion tensor imaging,' ensuring that any diffusion method, such as TBSS, was considered for inclusion. Upon reflection, we recognise that search terms containing ‘tractography’ and ‘PSMD’ are redundant, as these studies would also have been captured with the aforementioned general ‘diffusion’ search terms. Consequently, we removed them from our search terms. Furthermore, we have conducted a thorough search of the databases using specific search criteria for TBSS studies (using search terms: fatigue’ or ‘depression’ or ‘depressive symptoms’, in combination with ‘relapsing-remitting multiple sclerosis’ or ‘relapsing remitting multiple sclerosis’, in combination with ‘TBSS’ or ‘tract-based spatial statistics’) and conclude that no TBSS papers meeting our inclusion criteria were identified.

6. Why CTh and VBM atrophy patterns were not considered?

We apologise for having been unclear. Cortical thickness (CTh) and voxel-based morphometry (VBM) were certainly considered and not excluded from the review. Of the seventeen studies exploring structural MRI that were included in this review, however, only two described CTh and none used VBM. We therefore conclude that no VBM and limited CTh studies have been conducted within the context of our inclusion criteria. We have now added MRI measures to Table 3, to make it clear to the reader which studies investigated CTh.

---

## [Decision Letter · Decision Letter 1]

3 Jan 2024

PONE-D-23-09587R1Brain connectivity changes underlying depression and fatigue in relapsing-remitting multiple sclerosis: a systematic reviewPLOS ONE

Dear Dr. Kampaite,

Thank you for submitting your manuscript to PLOS ONE. After careful consideration, we feel that it has merit but does not fully meet PLOS ONE’s publication criteria as it currently stands. Therefore, we invite you to submit a revised version of the manuscript that addresses the points raised during the review process. Considering the important topic and potential impact of the manuscript I decided to let an additional reviewer to read the contribution, and based on that I decided that the manuscript needs a minor revision.

We look forward to receiving your revised manuscript.

Kind regards,

Peter Lundberg

Academic Editor

PLOS ONE

Journal Requirements:

Reviewers' comments:

Reviewer's Responses to Questions

**Comments to the Author**

1. If the authors have adequately addressed your comments raised in a previous round of review and you feel that this manuscript is now acceptable for publication, you may indicate that here to bypass the “Comments to the Author” section, enter your conflict of interest statement in the “Confidential to Editor” section, and submit your "Accept" recommendation.

Reviewer #3: (No Response)

2. Is the manuscript technically sound, and do the data support the conclusions?

Reviewer #3: Yes

3. Has the statistical analysis been performed appropriately and rigorously? 

Reviewer #3: N/A

4. Have the authors made all data underlying the findings in their manuscript fully available?

Reviewer #3: No

5. Is the manuscript presented in an intelligible fashion and written in standard English?

Reviewer #3: (No Response)

6. Review Comments to the Author

Reviewer #3: Thank you for letting me be a referee for this interesting manuscript. It contains important information and I think it should be published. I also find that the previous reviewers have addressed important issues, and the authors have responded and revised their text accordingly.

However, I miss important information on heat sensitivity and MS in the back ground - this is a typical feature of MS and is intertwined with fatigue in particular. The potential of heat sensitivity to increase fatigue as well as other core symptoms is generally overlooked Consequently, cryotherapy against fatigue should me mentioned more clearly, for exmaple in the introductory text on therapies used for fatigue. Suitable references by Flensner et al, especially https://doi-org.ezproxy.its.uu.se/10.1186/1471-2377-11-27

I also miss one paper on fMRI and fatigue that seems to fit with the inclusion criteria of the authors, i.e. dividing subtypes and reporting fatigue, available on Pubmed: Engstrom et al https://doi-org.ezproxy.its.uu.se/10.1002/brb3.181. This work is congruent with the works by Filippi, but also touches upon potential mechanisms of fatigue, which could be expanded a bit in the text.

Finally you may add the first collection of articles before selection using your criteria, appendix or supplement.

I will be happy to read a revised version.

7. PLOS authors have the option to publish the peer review history of their article (what does this mean?). If published, this will include your full peer review and any attached files.

Reviewer #3: **Yes: **Anne-Marie Landtblom

---

## [Author Response · Author response to Decision Letter 1]

25 Jan 2024

Journal Requirements:

We have carefully reviewed the reference list and found no inclusion of retracted papers. We did, however, observe several required modifications. We apologise for previously having overlooked these errors in our reference list. We have now made these modifications to the reference list where needed: 

1. Cao et al. - replaced correction with the original corrected paper. 

Cao Y, Diao W, Tian F, Zhang F, He L, Long X, et al. Gray Matter Atrophy in the Cortico-Striatal-Thalamic Network and Sensorimotor Network in Relapsing–Remitting and Primary Progressive Multiple Sclerosis. Neuropsychology Review. 2021;31(4):703-20. doi: 10.1007/s11065-021-09479-3.

2. Golde et al. – corrected duplicate.

Golde S, Heine J, Pöttgen J, Mantwill M, Lau S, Wingenfeld K, et al. Distinct Functional Connectivity Signatures of Impaired Social Cognition in Multiple Sclerosis. Frontiers in Neurology. 2020;11. doi: 10.3389/fneur.2020.00507.

3. Baker and Tadi – edited citation from ‘book’ to ‘website’.

Banker L, Tadi P. Neuroanatomy, Precentral Gyrus. In: StatPearls [Internet]. StatPearls Publishing, Treasure Island (FL); 2023 [updated 2023 Jul 24]. Available from: https://www.ncbi.nlm.nih.gov/books/NBK544218/.

4. Ghandili – edited citation from ‘book’ to ‘website’.

Ghandili M, Munakomi S. Neuroanatomy, Putamen. In: StatPearls [Internet]. StatPearls Publishing, Treasure Island (FL); 2023 [updated 2023 Jan 30]. Available from: https://www.ncbi.nlm.nih.gov/books/NBK542170/.

5. Valentine et al. – edited doi. 

Valentine TR, Alschuler KN, Ehde DM, Kratz AL. Prevalence, co-occurrence, and trajectories of pain, fatigue, depression, and anxiety in the year following multiple sclerosis diagnosis. Multiple Sclerosis Journal. 2022;28(4):620-31. doi: 10.1177/13524585211023352. PubMed PMID: 34132141.

Reviewers' comments:

Reviewer #3: 

1. I miss important information on heat sensitivity and MS in the back ground - this is a typical feature of MS and is intertwined with fatigue in particular. The potential of heat sensitivity to increase fatigue as well as other core symptoms is generally overlooked Consequently, cryotherapy against fatigue should be mentioned more clearly, for example in the introductory text on therapies used for fatigue. Suitable references by Flensner et al, especially https://doi-org.ezproxy.its.uu.se/10.1186/1471-2377-11-27

We agree with the reviewer and have now revised the introduction and discussion to include heat sensitivity as contributing factor to fatigue in MS and cryotherapy as potential treatment for it. 

Introduction:

“Fatigue can appear spontaneously, or be brought on by a combination of internal or external factors, such as mental or physical activity, heat sensitivity, humidity, acute infection, and food ingestion [7, 36]. Commonly suggested primary mechanisms of fatigue in MS involve the immune system or damage to the CNS, such as inflammatory processes (e.g. cytokines), endocrine dysregulation, axonal loss, demyelination, as well as functional connectivity changes. [9, 37, 38]. This review will focus on structural damage of the CNS in the white (WM) and grey matter GM), specifically changes in structural and functional brain connectivity, as potential underlying mechanism of fatigue in pwRRMS.”

“Additionally, antidepressants, cognitive behavioural therapy [6] and cryotherapy [55] have had some success in reducing both depression and fatigue symptomatology in MS. Given the limited treatment success, underlying CNS changes of fatigue and depression in MS need to be elucidated, which may aid development of more effective targeted treatments for both symptoms in MS.”

Discussion:

“It may also indicate a possibility for shared treatments such as cognitive behavioural therapy, cryotherapy and balance and/or multicomponent exercise, both of which show promising results in CFS [248-250].”

2. I also miss one paper on fMRI and fatigue that seems to fit with the inclusion criteria of the authors, i.e. dividing subtypes and reporting fatigue, available on Pubmed: Engstrom et al https://doi-org.ezproxy.its.uu.se/10.1002/brb3.181.

We thank the reviewer for suggesting the above paper. We have reviewed the article by Engstrom et al. and noticed the study included 15 people with MS. This is below our set inclusion criteria of n ≥ 20. We realised we did not include this criterion in the abstract, which may have caused confusion. We have now added it to the abstract accordingly. 

“Searched databases were PubMed, Web-of-Science and Scopus. Inclusion criteria were: studied participants with RRMS (n ≥ 20; ≥ 18 years old) and differentiated between MS subtypes; published between 2001-01-01 and 2023-01-18; used fatigue and depression assessments validated for MS; included brain structural, functional magnetic resonance imaging (fMRI) or diffusion MRI (dMRI). Sixty studies met the criteria: 18 dMRI (15 fatigue, 5 depression) and 22 fMRI (20 fatigue, 5 depression) studies.”

3. This work is congruent with the works by Filippi, but also touches upon potential mechanisms of fatigue, which could be expanded a bit in the text.

We agree with the reviewer and have revised the introduction to expand on the potential primary mechanisms of fatigue in MS. 

“Fatigue can appear spontaneously, or be brought on by a combination of internal or external factors, such as mental or physical activity, heat sensitivity, humidity, acute infection, and food ingestion [7, 36]. Commonly suggested primary mechanisms of fatigue in MS involve the immune system or damage to the CNS, such as inflammatory processes (e.g., cytokines), endocrine dysregulation, axonal loss, demyelination, as well as functional connectivity changes [9, 37, 38]. This review will focus on structural damage of the CNS in the white (WM) and grey matter (GM), specifically changes in structural and functional brain connectivity, as potential underlying mechanism of fatigue in pwRRMS.”

4. Finally, you may add the first collection of articles before selection using your criteria, appendix or supplement.

We thank the reviewer for this excellent suggestion and have now created a list of all the articles included in our initial database search, with duplicates removed, and added this as supporting information file S11.

---

## [Editor Report · Decision Letter 2]

15 Feb 2024

Brain connectivity changes underlying depression and fatigue in relapsing-remitting multiple sclerosis: a systematic review

PONE-D-23-09587R2

Dear Dr. Kampaite,

We’re pleased to inform you that your manuscript has been judged scientifically suitable for publication and will be formally accepted for publication once it meets all outstanding technical requirements.

Kind regards,

Peter Lundberg

Academic Editor

PLOS ONE
---

## [Editor Report · Acceptance letter]

21 Mar 2024

PONE-D-23-09587R2 

PLOS ONE

Dear Dr. Kampaite, 

I'm pleased to inform you that your manuscript has been deemed suitable for publication in PLOS ONE. Congratulations! Your manuscript is now being handed over to our production team.

Kind regards, 

on behalf of

Professor Peter Lundberg 

Academic Editor

PLOS ONE